# Callyspongiolide kills cells by inducing mitochondrial dysfunction via cellular iron depletion

Jaeyoung Ha [1] & Seung Bum Park [1,2,3 ✉]

The highly cytotoxic marine natural product callyspongiolide holds great promise as a warhead of antibody-drug conjugate in cancer therapeutics; however, the mechanism underlying its cytotoxicity remains unclear. To elucidate how callyspongiolide kills cells, we employed label-free target identification with thermal stability-shift-based fluorescence difference in two-dimensional (2-D) gel electrophoresis (TS-FITGE), which allowed observation of a unique phenomenon of protein-spot separation on 2-D gels upon treatment with callyspongiolide at increasing temperatures. During our exploration of what proteins were associated with this phenomenon as well as why it happens, we found that callyspongiolide induces mitochondrial/lysosomal dysfunction and autophagy inhibition. Moreover, molecular biology studies revealed that callyspongiolide causes lysosomal dysfunction, which induces cellular iron depletion and leads to mitochondrial dysfunction and subsequent cytotoxicity. Notably, these effects were rescued through iron supplementation. Although our approach was unable to reveal the direct protein targets of callyspongiolide, unique phenomena observed only by TS-FITGE provided critical insight into the mechanism of action of callyspongiolide and specifically its cytotoxic activity via induction of mitochondrial dysfunction through cellular iron depletion caused by lysosomal deacidification, which occurred independent of known programmed cell death pathways.

[1] Department of Biophysics and Chemical Biology, Seoul National University, Seoul 08826, Korea. [2] CRI Center for Chemical Proteomics, Department of Chemistry, Seoul National University, Seoul 08826, Korea. [3] SPARK Biopharma, Inc, Seoul 08791, Korea. ✉email: sbpark@snu.ac.kr

A myriad of bioactive compounds and drug-like molecules are found in nature, some of which have been highlighted as therapeutic drug candidates owing to their unique structural properties[1,2]. Terrestrial plants and microbes are frequently traditional sources for these natural products; however, the marine environment has recently become an attractive source for new types of natural products[3]. With the development of techniques for collecting marine samples and chemical characterization, increasing numbers of marine natural products have been discovered, identified, and subjected to screening for drug discovery. In fact, hit rates for drug leads related to marine natural products are reportedly higher than those of terrestrial origin, and a majority of marine natural products exhibit cytotoxic effects against cancer cell lines[4–6].

The biological activities of such products are based on interactions between natural products and specific proteins; therefore, identification of direct binding targets should be undertaken to reveal the precise mechanisms associated with the actions of natural products in cells and organisms, as well as their subsequent development into therapeutic agents[7]. For target identification (ID), conventional methods require structure-activity relationship (SAR) studies in order to incorporate additional functional handles on the compounds to create target ID probes allowing their immobilization on solid supports and enabling the enrichment of target proteins via affinity-based pulldown. However, the process of labeling bioactive natural products has represented a major bottleneck due to their limited quantity in nature, resulting in difficult procurement (several milligrams of these compounds from hundreds of grams of the organisms), as well as their complex structures, which makes SAR studies difficult. Recently, new target ID methods (i.e., label-free target identification) that do not require compound modifications were developed to broaden the applicable subjects for target ID by utilizing the intrinsic properties of the protein (e.g., heat-, chemical-, and proteolysis-induced denaturation/degradation) that can be altered upon ligand binding[8]. Among these methods, thermal stability shift-based label-free target ID methods, including cellular thermal shift assay (CETSA)[9], thermal proteome profiling (TPP)[10], and thermal stability shift-based fluorescence difference in two-dimensional (2-D) gel electrophoresis (TS-FITGE)[11], were developed and applied for the identification of direct binding partners of many bioactive small molecules and natural products[12,13]. However, there remains a possibility that heat-based methods cannot provide information regarding protein targets when the thermal stability shifts of proteins are unaltered upon ligand interaction.

Callyspongiolide (CSG) (Fig. 1) is a natural compound derived from marine sponges and first discovered in 2013[14]. Apart from its complex molecular structure, researchers observed a strong cytotoxic effect against human T and B lymphoma cell lines[14]. Additionally, organic chemists have been attracted to this marine macrolide due to its unique ene-yne-ene lateral chain and bromophenol moiety, as well as its highly potent cytotoxicity. Two years after its initial discovery, researchers successfully synthesized CSG and its diastereomers, assigned exact stereochemistries, and confirmed their antiproliferative effects in various human cancer cell lines from different histological origins[15]. Moreover, Manoni and Harran[16] confirmed the cytotoxic effect in human T lymphocytes and reported its cytotoxicity in a yeast bearing mutation of the PDR5 drug transporter. Further evaluation revealed a cell death mechanism in which the canonical apoptosis pathway was excluded, as cell death caused by CSG was not blocked by pan-caspase inhibitors[14,16], with vacuolar-type ATPase suggested as a potent target of CSG in yeast[17]. However, the precise molecular mechanism by which this marine natural compound exerts its cytotoxicity remains unclear. Here, we demonstrated CSG cytotoxicity in human cancer cell lines and revealed the molecular mechanism associated with its killing of human cancer cells using TS-FITGE in an unconventional way.

## Results

**The natural form of callyspongiolide harbors greater cytotoxicity than its synthetic epimer.** After total synthesis of CSG and its synthetic diastereomers, Ye et al.[15] reported that the natural form of CSG [(−)-callyspongiolide] and its synthetic epimer at C-21, epi-CSG [21-epi-(−)-callyspongiolide] (Fig. 1) exhibited potent cytotoxicity in double- or triple-digit nanomolar potency of the half-maximal inhibitory concentration ($IC_{50}$). Interestingly, they claimed that synthetic epimer (epi-CSG) was more potent than the natural form of CSG against most cancer cell lines tested (MCF7, SH-SY5Y, HeLa, HT-29, RKO, and PC-3), whereas the opposite was observed in H1299 and Jurkat cells.

Prior to investigation of the CSG mechanism of action, we confirmed the difference in cytotoxicity between CSG and epi-CSG according to cell lines by performing MTT cell-viability assays in nine human cancer cell lines treated with CSG or epi-CSG. Treatment of cells with either form for 24 h induced no cytotoxic effect; however, cytotoxicity gradually increased in 48- and 72-h treatments (Supplementary Fig. 1). Interestingly, we observed that CSG had a double-digit nanomolar $IC_{50}$ (19.3–97.8 nM), whereas epi-CSG exhibited significantly lower potency, with a triple-digit nanomolar $IC_{50}$ (184–722 nM) in all tested cell lines (Table 1). In contrast to previously reported observations, the stereochemical configuration of CSG at C-21 was crucial for its cytotoxicity in human cancer cell lines. Therefore, we selected CSG for further elucidation of the mechanism underlying its cytotoxicity.

**CSG kills cells in an apoptosis- and parthanatos-independent manner.** We first conducted immunoblot analyses of caspase-3, caspase-9, and poly(ADP-ribose) polymerase (PARP), as well as performed Annexin V and propidium iodide staining to investigate the involvement of the apoptotic pathway in the cytotoxic effect of CSG. During canonical apoptosis, caspase-3, caspase-9, and PARP are cleaved, and Annexin V staining increases upon treatment with the apoptosis-inducing compound etoposide

**Fig. 1 Chemical structures of CSG and epi-CSG.** The chemical structures of natural form of callyspongiolide [(−)-callyspongiolide, CSG] and its synthetic epimer at C-21 [21-epi-(−)-callyspongiolide, epi-CSG].

| Table 1 IC₅₀ (nM) values at 72 h of CSG and *epi*-CSG in human cancer cell lines. | | | |
|---|---|---|---|
| | | IC50 at 72 h (nM) | |
| Origin | Cell line | CSG | *epi*-CSG |
| Lung | A549 | 64.8 ± 1.1 | 467± 23 |
| Colon | HCT116 | 60.4 ± 1.1 | 392 ± 64 |
| Kidney | HEK293T | 97.8 ± 9.5 | 722 ± 32 |
| Cervix | HeLa | 70.6 ± 5.3 | 530 ± 33 |
| Liver | HepG2 | 35.5 ± 1.9 | 316 ± 17 |
| T lymphocyte | Jurkat | 34.7 ± 1.7 | 296 ± 3.1 |
| Breast | MCF7 | 34.8 ± 4.9 | 275 ± 22 |
| Prostate | PC3 | 19.3 ± 3.7 | 184 ± 14 |
| Bone marrow | SH-SY5Y | 81.1 ± 3.9 | 646 ± 21 |

(Supplementary Fig. 2a–c)[18]. However, CSG did not induce cleavage of caspase-3 and caspase-9, which was consistent with previous studies showing that CSG-mediated cell death was not diminished by pan-caspase inhibitor treatment[14,16]. Nevertheless, we detected slight PARP cleavage along with an increase in Annexin V staining upon CSG treatment, indicating that its cytotoxic activity might be mediated by a caspase-independent, PARP-dependent pathway.

Parthanatos is a caspase-independent programmed cell death pathway primarily mediated by translocation of the apoptosis-inducing factor (AIF) protein from mitochondria into the nucleus, followed by DNA condensation and fragmentation[19,20]. Because the increases in PARP cleavage and Annexin V staining characteristic of the parthanatos pathway were similar to what we observed following CSG treatment, we explored whether CSG-mediated cell death was associated with parthanatos by assessing AIF translocation and DNA fragmentation. However, AIF did not translocate into the nucleus upon CSG treatment, as confirmed by immunofluorescence. Instead, AIF co-localized with mitochondrial membrane protein translocase of outer mitochondrial membrane (TOMM)20, regardless of CSG treatment (Supplementary Fig. 2d). Moreover, DNA fragmentation monitored via TUNEL assay indicated that DNA remained intact, even in the presence of CSG, as opposed to following DNase treatment (Supplementary Fig. 2e). These findings suggested that despite CSG-mediated induction of increased Annexin V staining and minor PARP cleavage, cell death was mediated by neither canonical apoptosis nor parthanatos.

**Label-free target identification of CSG**. To elucidate the mechanism underlying CSG cytotoxicity, we performed target identification in order to obtain a list of proteins directly engaged and functionally regulated by CSG. However, conventional target ID methods require affinity-based probes via structural modification of active compounds, which is difficult in the case of CSG due to its complex structure. Therefore, we employed a label-free target ID method. We previously reported the development and application of such a method (TS-FITGE) based on protein thermal stability shift upon ligand engagement[11,21,22]. In the present study, we employed TS-FITGE for CSG target ID. Briefly, A549 lung cancer cells were treated with dimethyl sulfoxide (DMSO) or CSG and then heat shocked for 3 min from 44 °C to 64 °C at 2 °C intervals. The resulting cells were then lysed and centrifuged to obtain the soluble protein fractions, after which those from DMSO- or CSG-treated cells were chemically conjugated with Cy3 and Cy5 fluorescent dyes, respectively. Proteins from each group at the same temperature were mixed and resolved by 2-D gel electrophoresis, and the gels were analyzed via fluorescent scanning (Supplementary Fig. 3). As the temperature increased, some yellow spots on the 37 °C gel were split in two, side by side. These then turned into green and red

spots, respectively (Fig. 2a–e). This phenomenon was also observed in other TS-FITGE experiments using HeLa human cervical cancer cells and Jurkat human T lymphocytes (Supplementary Fig. 4). Based on a set of experimental replicates, we selected 10 pairs of green-red protein spots (noted as 1G/1R to 10G/10R in Fig. 2b–e), and spot authenticity was confirmed via fluorescence intensity quantification of each protein spot at 60 °C and delineated by the Cy5:Cy3 ratio on a log scale (Fig. 2f). Proteins in spots 1G/1R to 10G/10R were analyzed by tandem mass spectrometry and identified as succinate dehydrogenase complex flavoprotein subunit A (SDHA), AIF, ATP synthase subunit α (ATP5F1A), ubiquinol-cytochrome c oxidoreductase subunit 2 (UQCRC2), TOMM40, catalytic subunit of protein phosphatase 2A (PP2AC), ATP synthase B chain (ATP5F1), NADH:ubiquinone oxidoreductase subunit B10 (NDUFB10), and basic transcription factor 3 (BTF3), corresponding to spot pairs 1 through 10, respectively (Table S1 and Supplementary Fig. 5). These results indicated that proteins in the side-by-side green-red spot pairs were the same.

In fact, the outcome of label-free target ID with CSG differed from our other TS-FITGE studies, wherein specific protein spots usually became reddish on the 2-D gel as the temperature increased due to the thermal stabilization of target proteins[21,22]. The direct engagement of bioactive compounds with the resulting target-protein candidates can be verified via CETSA[9]. The thermal stability of most proteins from spots 1G/1R to 10G/10R did not shift following CSG treatment (Supplementary Fig. 6), which might be due to the fact that the same proteins in separated green-red spots on 2-D gels were merged into a single band in one-dimensional gels. Slight thermal destabilization of PP2AC was observed, but CSG-mediated cytotoxicity was unaffected by PP2AC knockdown in the cells (Supplementary Fig. 7).

We inferred that green-red spot pairs on 2-D gels at high temperature might not be caused by direct binding of CSG to each protein. Instead, this event could be induced by the change of cellular environment, especially within certain organelles. We then performed TS-FITGE with an A549 cell lysate, in which cellular compartments had been disrupted (Supplementary Fig. 8). Interestingly, we did not observe any green-red spot pairs at high temperature, implying that the unique phenomenon of green-red spot splitting on 2-D gels was dependent on the intracellular microenvironment. Furthermore, no other protein spots became red or green, indicating that its protein binding partners might be difficult to study via label-free target ID methods using thermal stability shift.

**Blockade of protein-spot migration upon treatment with CSG and a proton-gradient uncoupler**. Although we failed to identify the direct protein targets of CSG via TS-FITGE, we revealed that CSG induces protein-spot separation on 2-D gels at high temperature, which was greatly influenced by cellular compartmentalization. Hence, we conducted network analysis of the proteins obtained from TS-FITGE experiments using STRING[23] in order to explore which protein groups are affected by the disturbance of cellular environments upon treatment with CSG. As shown in Supplementary Fig. 9a, most of the proteins obtained via TS-FITGE were associated with the mitochondria, and five proteins among those belonged to the mitochondrial electron-transport system. Based on these results, we questioned whether CSG affects mitochondrial morphology and respiratory function.

Mitochondrial morphology can be monitored via live-cell fluorescent imaging with Mitotracker, a fluorescent dye that selectively stains mitochondria. The thread-like mitochondrial structures of cells were fragmented upon 12-h CSG treatment and

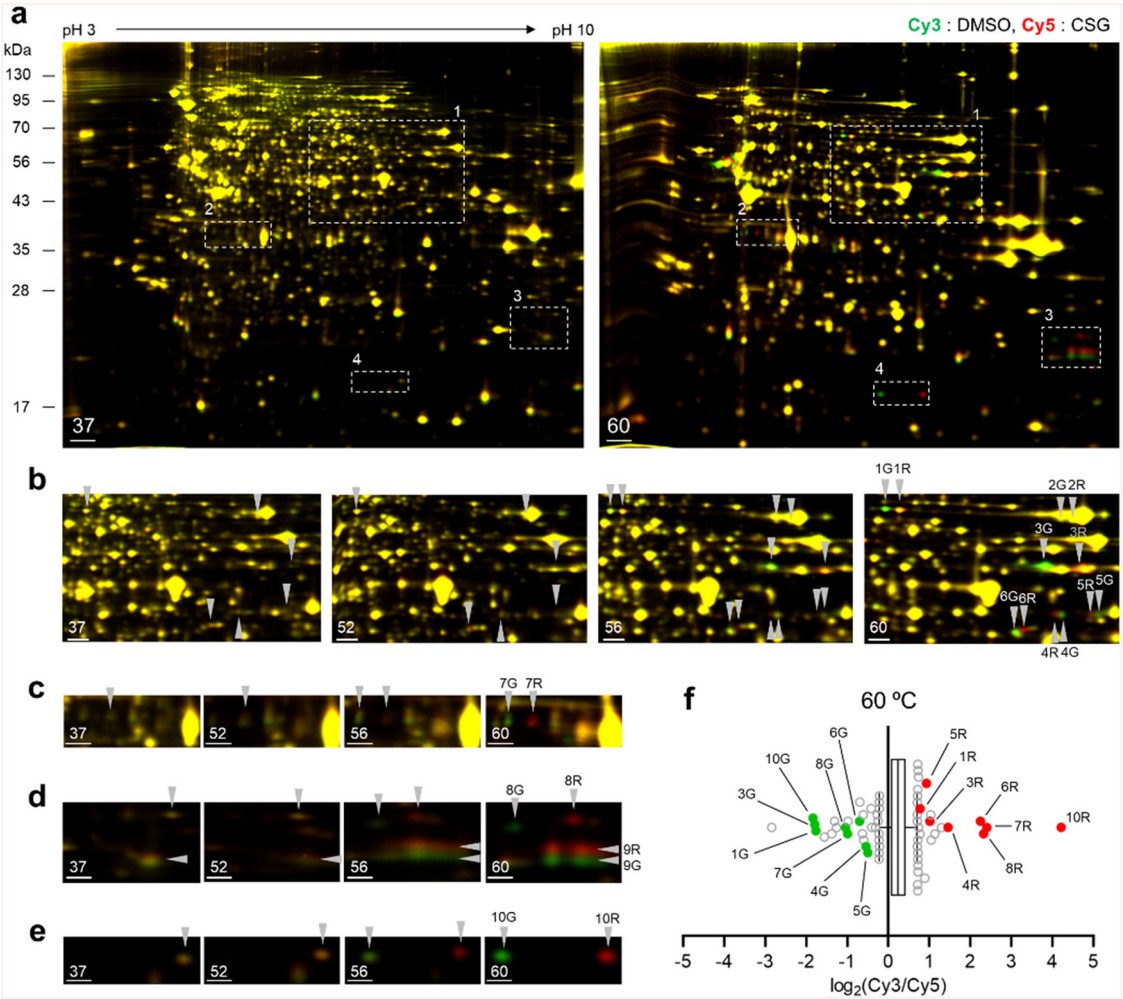

**Fig. 2 Label-free target ID of CSG using TS-FITGE. a** Representative images of TS-FITGE (pH 3–10) with CSG at 37 and 60 °C in A549 cells. Overlaid images of the Cy3 channel (green, DMSO-treated proteome) and Cy5 channel (red, CSG-treated proteome). Regions 1–4 in the white-dotted lines are magnified in (**b–e**). Images at all temperature range are available in Supplementary Fig. 3. **a–e** Magnified images of regions 1–4 noted in (**a**), respectively, at 37, 52, 56, and 60 °C. **f** Box and whisker plot showing the distribution of the Cy5/Cy3 fluorescence intensity ratio for each spot at 60 °C. The whiskers indicate 2.5–97.5 percentiles. The spots denoted in (**b–e**) are indicated as green and red dots. Scale bar, 1 cm (**a**); 0.5 cm (**b, c**); 0.25 cm (**d, e**).

remained so until cell death (Supplementary Fig. 9b–c). Thereafter, mitochondrial respiratory function was assessed using the Seahorse bioenergetic analyzer, which allows measurement of cellular oxygen consumption rate (OCR) following sequential treatments with mitochondrial respiration inhibitors. As shown in Supplementary Fig. 9d, CSG caused a gradual time-dependent decrease in the OCR, whereas the OCR was unaffected by inhibitors following 24-h CSG treatment, implying that CSG impairs mitochondrial respiration. However, the total number of mitochondria was not significantly affected by CSG treatment (Supplementary Fig. 9e).

We then conducted TS-FITGE experiments following pretreatments with a mitochondrial respiration inhibitor in both DMSO- and CSG-treated groups, as CSG-mediated disruptions of mitochondrial morphology and respiratory function might induce environmental changes along with the functional and structural changes of certain mitochondrial proteins, thereby leading to unique protein-spot separation at increasing temperatures on 2-D gels. TS-FITGE performed with CSG experiments following pretreatment with specific complex I, III, and V inhibitors [rotenone (Rot), antimycin A (AA), and oligomycin A (Oligo), respectively] revealed the continued existence of green-red spot pairs at high temperature (Supplementary Fig. 10a–d), indicating that protein spot separation did not occur through perturbation of the mitochondrial respiratory

function even though most of the proteins obtained from TS-FITGE experiments belong to the electron-transport chain complex. However, when cells were pretreated with trifluoromethoxycarbonylcyanide phenylhydrazone (FCCP), which disrupts the proton gradient across the mitochondrial inner membrane, green-red protein-spot separation disappeared, and the 2-D gel pattern was similar to that observed at 37 °C, as if protein-spot separation did not occur, even at high temperature (Fig. 3a–c and Supplementary Fig. 10e). Interestingly, when we performed TS-FITGE with FCCP, the same phenomenon of protein-spot separation at high temperatures occurred (Fig. 3d and Supplementary Fig. 11). This observation implied that FCCP treatment phenocopied the TS-FITGE outcomes for CSG. These findings suggested that as the temperature increased, some protein spots migrated to either the right or left side of 2-D gels, whereas CSG and FCCP blocked protein-spot migration, resulting in the green-red protein-spot separation observed upon merging of Cy3 and Cy5 images (Fig. 3e). Therefore, the molecular mechanism of CSG-mediated cytotoxicity could be unraveled by tracing how FCCP works in cells.

**CSG impairs lysosome acidity rather than mitochondrial potential, leading to autophagy inhibition.** FCCP is a well-known mitochondrial potential uncoupler[24], but it also disrupts the

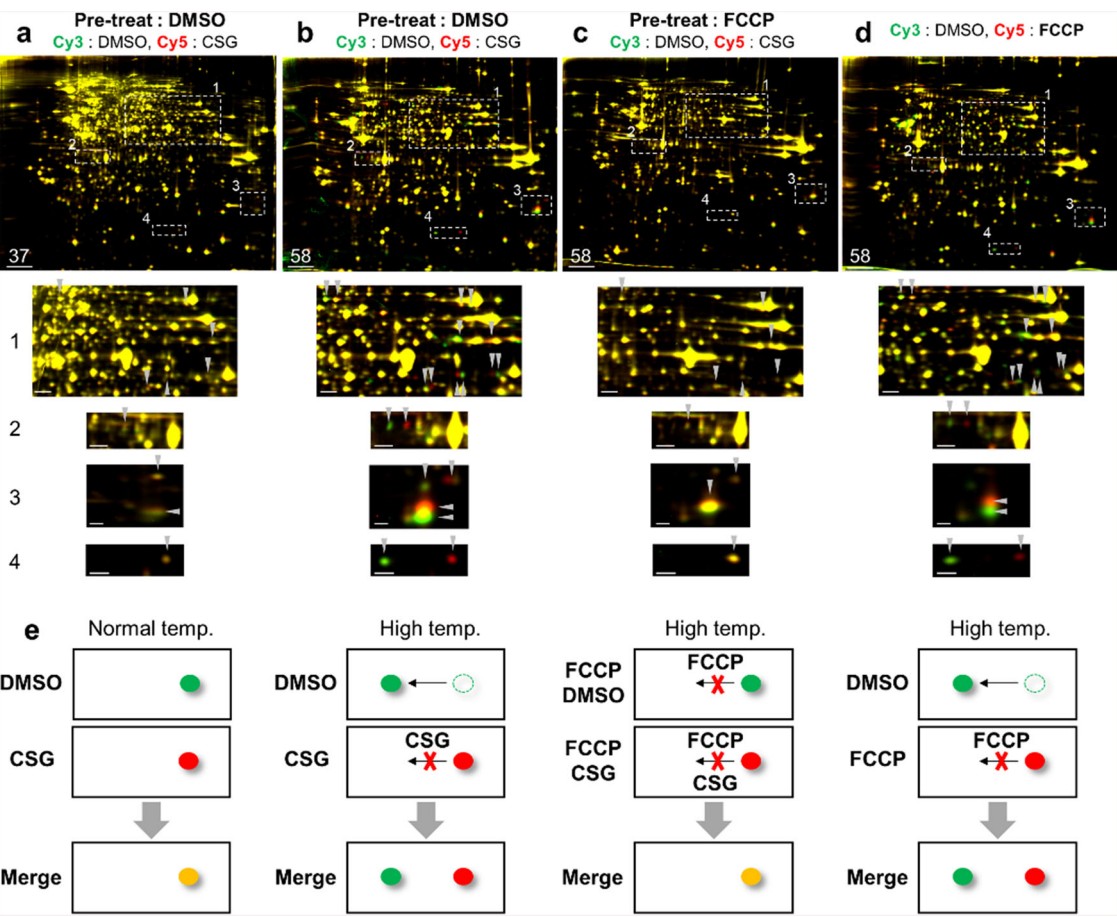

**Fig. 3 Protein-spot migration on 2-D cells at high temperatures was blocked by CSG or the proton-gradient uncoupler FCCP. a–c** Representative images of TS-FITGE (pH 3–10) with CSG following pretreatment with DMSO or FCCP at 37 or 58 °C. Overlaid images of the Cy3 channel (green, DMSO-treated proteome) and Cy5 channel (red, CSG-treated proteome). Regions 1–4 in the white-dotted line are magnified below. Images at all temperature ranges are available in Supplementary Fig. 10a and e. **d** Representative images of TS-FITGE (pH 3–10) with FCCP at 58 °C. Overlaid images of the Cy3 channel (DMSO-treated proteome) and Cy5 channel (FCCP-treated proteome). Regions (1–4) in the white-dotted lines are magnified below. Images at all temperature ranges are available in Supplementary Fig. 11. **e** Simplified scheme of protein-spot migration on 2-D gels at high temperatures and blockade by CSG or FCCP. Scale bar, 2 cm (top row); 0.5 cm (Regions 1, 2); 0.25 cm (Regions 3, 4).

proton gradient in lysosomes[25], where protons are continuously and actively pumped in to maintain the acidic microenvironment[26]. Based on TS-FITGE data, we confirmed that FCCP phenocopied the TS-FITGE outcome of CSG; therefore, we questioned whether CSG would also affect mitochondrial membrane potential and lysosome acidity. To address this question, we performed live-cell fluorescent imaging with tetramethylrhodamine ethyl ester (TMRE), which specifically stains mitochondria with active membrane potential, and Lysotracker, which stains the acidic lysosome[27,28]. FCCP disrupted both mitochondrial membrane potential and lysosome acidity (Fig. 4a, b); however, although CSG did not affect mitochondrial membrane potential, it did impair lysosome acidity, as observed following treatment with bafilomycin A1 (Baf), a well-known inhibitor of vacuolar-type ATPase[29]. Vacuolar-type ATPase has a pivotal role for lysosomal acidification, hence, we also wondered whether vacuolar-type ATPase would be affected by CSG. The catalytic activity of vacuolar-type ATPase was monitored by measuring fluorescence intensity of Oregon Green 514-dextran-loaded lysosome in vitro, which decreased upon re-acidification of the lysosome followed by activation of vacuolar-type ATPase with ATP and $MgCl_2$ (Fig. 4c)[30]. However, CSG treatment did not allow lysosomal acidification like Baf treatment, meaning that CSG also inhibits vacuolar-type ATPase, which was consistent with a recent report suggesting CSG as a potent inhibitor of

vacuolar-type ATPase in yeast[17]. Therefore, we concluded that CSG disrupts the proton gradient across the lysosomal membrane by inhibiting vacuolar-type ATPase, leading to impaired lysosome acidification.

Given that CSG impairs lysosomal acidification, we speculated that the blockade of protein-spot migration in TS-FITGE experiments was associated with lysosome neutralization. Thus, we conducted TS-FITGE with Baf or chloroquine (CQ), another lysosomal neutralizer[31] (Supplementary Fig. 12). Interestingly, we observed the same green-red spot separation on 2-D gels at high temperatures, proving that the spot-separation phenomenon on TS-FITGE was due to the CSG-mediated alteration of lysosomal pH by inducing lysosomal neutralization. We then hypothesized that CSG would target the same target of Baf that induces lysosomal neutralization. Vacuolar ATPase $V_0$ complex C subunit (ATP6V0C), a subunit of the $V_0$ complex of vacuolar-type ATPase embedded in the lysosomal membrane, was previously identified as a functional target of Baf[32]. Hence, we performed CETSA of ATP6V0C upon treatment with either CSG or Baf; however, we observed no shift in the thermal stability of ATP6V0C (Supplementary Fig. 13). This indicated the reason why TS-FITGE was unable to identify this putative target protein.

Poor lysosomal acidification is associated with suppression of the autophagic pathway[33]. To investigate whether CSG influences

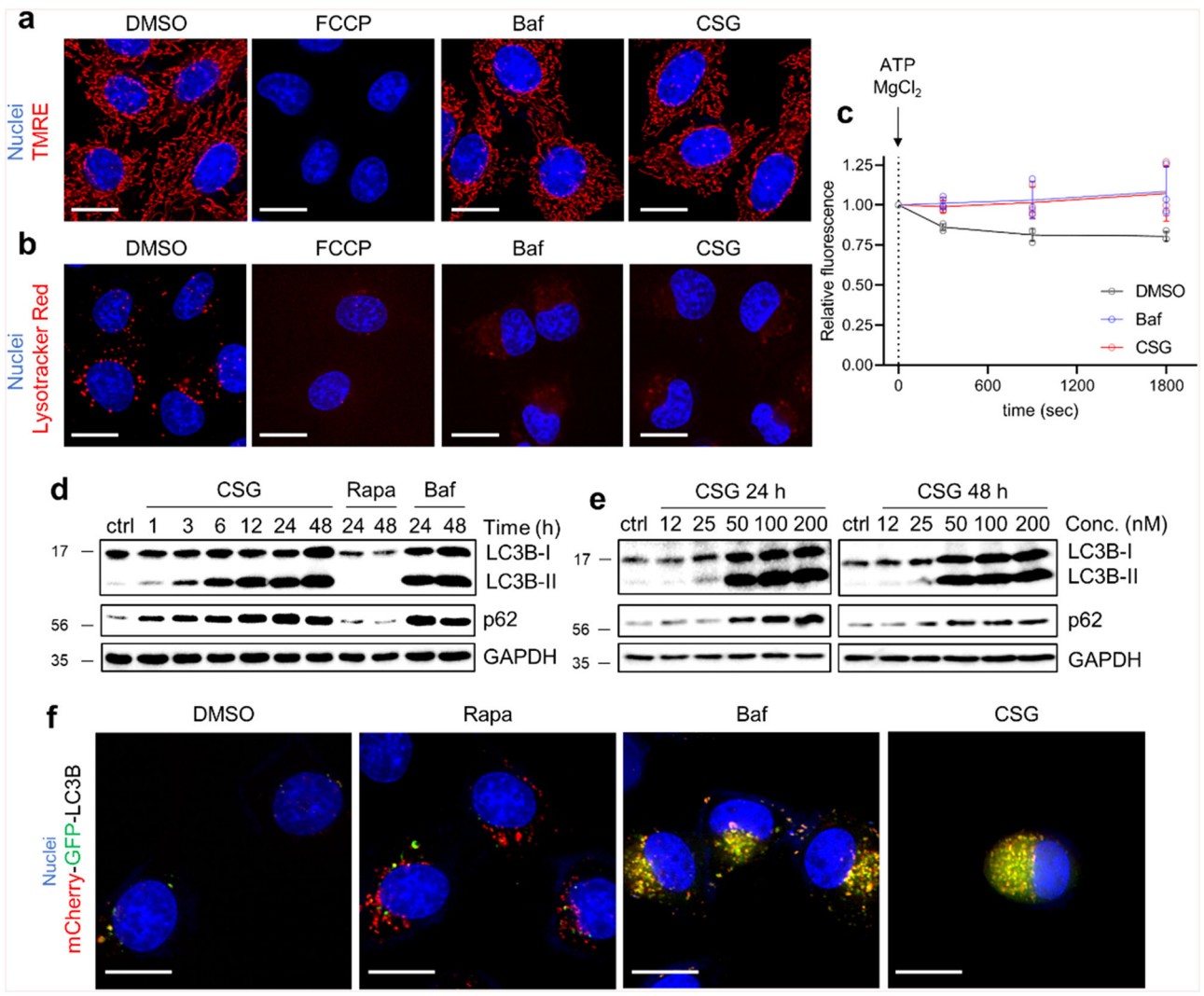

**Fig. 4 CSG impairs lysosome acidity, but not mitochondrial potential, leading to autophagy inhibition. a, b** Representative live-cell fluorescence imaging of mitochondria with active potentials using TMRE (**a**) and acidic lysosomes using Lysotracker (**b**) following treatment of A549 cells with FCCP (20 μM), Baf (20 nM), or CSG (200 nM) for 3 h. Nuclei were stained with Hoechst 33342. Scale bar, 10 μm. **c** in vitro vacuolar-type ATPase activity assay. To dextran-OG514-loaded organellar fraction containing lysosomal membrane isolated from HEK293T cells, Baf (20 nM) or CSG (200 nM) were added with $Na_2ATP$ (5 mM) and $MgCl_2$ (5 mM). Data represent the mean ± SD ($n = 3$). See "Methods" for details. **d, e** Immunoblotting of LC3B and p62 following treatment of A549 cells with CSG (200 nM), Rapa (500 nM), or Baf (20 nM) for the indicated times (**d**). The indicated CSG concentration was applied to A549 cells for 24 h or 48 h (**e**). **f** Representative live-cell fluorescent images of A549 cells transfected with the mCherry-GFP-LC3B plasmid following treatment with Rapa (500 nM), Baf (20 nM), or CSG (200 nM) for 24 h. Scale bar, 10 μm.

autophagy, we first conducted immunoblotting of light-chain 3B (LC3B) and p62. As shown in Fig. 4d and e, CSG induced the conversion of LC3B, as well as p62 accumulation, in a time- and dose-dependent manner, indicative of CSG-mediated autophagic flux inhibition due to lysosomal deacidification and similar to the effects of Baf. We then performed live-cell imaging to monitor mCherry-GFP-LC3B[34] in order to confirm that CSG inhibits autophagic flux (Fig. 4f). When autophagy is induced by rapamycin (Rapa), autophagolysosomes appear as red puncta due to loss of the GFP signal in the acidic environment. However, when autophagy is blocked via Baf, yellow puncta accumulate, as the GFP signal is retained within non-acidic autophagosomes. Confirming the immunoblot results, CSG inhibited autophagic flux similar to Baf, as indicated by the accumulation of yellow puncta observed in mCherry-GFP-LC3B imaging.

We then questioned whether autophagy inhibition was involved in CSG-induced cell death as suggested by the early onset of LC3B conversion. Following the knockdown of

autophagy-related 5 (ATG5), a protein crucial for the propagation of phagophores into mature autophagosomes[35], using small-interfering RNA (siRNA), we determined whether CSG-mediated cytotoxicity could be attributed to the accumulation of dysfunctional autophagolysosomes by treating cells with a range of CSG concentrations. As shown in Supplementary Fig. 14, CSG was slightly less toxic under ATG5 knockdown, even though autophagy inhibition did not fully account for CSG-mediated cell death.

**CSG induces mitochondrial dysfunction via cellular iron depletion, leading to cell death**. We hypothesized that lysosomal neutralization could be a key factor in CSG-induced cell death, given that autophagic cell death was excluded as an underlying mechanism. The mechanism driving cell death induced by compounds that cause lysosome neutralization has remained controversial. Some studies suggest that Baf-, CQ-, and

concanamycin A-induced cell death is mediated by apoptosis accompanied by DNA fragmentation[36–41], endoplasmic reticulum (ER) stress[42], autophagic vacuole accumulation[43], and caspase-independent mitogen-activation protein kinase (MAPK) pathway activation[44]. Although there have been a series of conflicting reports on this matter, a recent study showed that disturbing cholesterol biosynthesis and cellular iron homeostasis in cells sensitized them to ammonium base- or Baf-induced death via lysosomal pH alteration[45]. These findings implied that cellular cholesterol or iron depletion might be involved in cytotoxicity mediated via lysosomal neutralization. Thus, we examined whether cholesterol or iron supplementation can rescue CSG-induced cell death. Indeed, iron supplementation with ferric citrate, but not cholesterol or sodium citrate supplementation, prevented cell death caused by CSG (Fig. 5a). Similarly, cellular iron deprivation by the iron chelator deferoxamine (DFO) also showed gradual cytotoxicity over time, and the cells were rescued by ferric citrate supplementation, not by sodium citrate (Supplementary Fig. 15). These results indicate that cellular iron depletion underlies CSG-mediated cytotoxicity.

When cellular iron is sufficient, it is mainly stored in ferritin proteins within the cell[46]. By contrast, iron deficiency results in the incorporation of the iron–ferritin complex into autophagosomes, followed by their fusion with lysosomes, and the resulting release of iron from ferritin via ferritinophagy in an autophagy-related process[47,48]. Ferritinophagy replenishes cellular iron and supplies it to mitochondria, where heme and iron–sulfur clusters are synthesized and utilized as cofactors for proteins in the electron-transport chain[49]. Acidic lysosomes and intact autophagy are prerequisites for ferritinophagy. By contrast, lysosomal dysfunction and autophagy inhibition lead to cellular iron depletion, followed by disruption of mitochondrial respiration[50]. Based on these findings, we questioned whether CSG depletes cellular iron, leading to mitochondrial dysfunction. To this end, we monitored cellular iron via live-cell imaging with the fluorescent iron probe FerroOrange (Fig. 5b, c, and Supplementary Fig. 16). FerroOrange fluorescence was enhanced as cellular iron was replenished via ferric citrate supplementation, but diminished as a result of cellular iron depletion using DFO without cytotoxicity for 24-h treatment (Supplementary Fig. 15). When the cells were treated with CSG, the fluorescence intensity gradually decreased over time, proving that cellular iron was indeed depleted by CSG. We next investigated other cellular responses to iron deficiency caused by CSG. Ferritin heavy chain (FTH1) protein and cellular heme are correlated to bioavailable cellular iron, hence, FTH1 and cellular heme level decreased by iron depletion upon CSG or DFO treatment (Supplementary Fig. 17). Another marker of cellular iron levels is the amount of *transferrin receptor* (TFRC) mRNA, which encodes the receptor required for endocytosis of transferrin complexed with ferric iron[51]. *TFRC* mRNA is degraded when cellular iron is sufficient, and stabilized as a result of cellular iron deficiency. We observed that *TFRC* mRNA level decreased following ferric citrate supplementation (Fig. 5d), whereas CSG or DFO treatment enhanced *TFRC* mRNA levels, an effect which was attenuated upon iron supplementation. Its corresponding protein TfR1 also showed a similar tendency to the mRNA level (Supplementary Fig. 17a) with less extent. These observations confirmed that CSG induced cellular iron depletion, which could be reversed through iron supplementation.

We then explored the impact of cellular iron depletion by CSG on mitochondria. As noted, iron–sulfur clusters are required for the function of some proteins in the electron-transport chain, such as NADH:ubiquinone oxidoreductase 75 kDa (NDUFS1), succinate dehydrogenase iron–sulfur subunit (SDHB), and ubiquinol-cytochrome c oxidoreductase iron–sulfur subunit (UQCRFS1). Upon depletion of cellular iron by CSG or DFO, the stability of those proteins was compromised, but the effect was reversed through iron supplementation (Fig. 5e). Especially, NDUFS1 and UQCRFS1 have a key role in the assembly of mitochondrial respiratory complex[52–55], hence, such destabilization of the proteins upon CSG or DFO treatment could lead to the impaired mitochondrial respiration. Thus, we questioned whether the TS-FITGE protein-spot-separation phenomenon caused by CSG might result from protein instability upon cellular iron depletion after lysosomal neutralization. Hence, we conducted the TS-FITGE experiment with DFO along with ferric citrate co-treatment for TS-FITGE with CSG (Supplementary Fig. 18). Iron depletion via DFO did not result in protein-spot separation on 2-D gels at high temperature, and CSG-mediated spot separation was not reversed through iron supplementation, indicating that cellular iron was irrelevant to the protein-spot-separation event.

We then hypothesized that the CSG-induced mitochondrial changes observed in this study could arise from the instability of proteins in the electron-transport chain caused by cellular iron depletion. Therefore, we questioned whether mitochondrial changes could be rescued through iron supplementation. We found that fragmented mitochondrial structure and dysregulated mitochondrial respiration were reversed to normal states via ferric citrate supplementation, but not via sodium citrate supplementation, meaning that cellular iron is crucial for mitochondrial homeostasis (Fig. 5f–h). Impairment of mitochondrial respiration generates excess reactive oxygen species (ROS)[56]. CSG induced an increase in cellular and mitochondrial ROS, as indicated by the fluorescent ROS probes 2′,7′-dichlorofluorescein diacetate (DCFDA) and MitoSOX, respectively (Fig. 5i). The increased cellular and mitochondrial ROS levels were also reversed by supplementation of ferric citrate. Because elevated cellular ROS can directly induce cell death as well as ROS-mediated ferroptosis[57], we examined whether CSG cytotoxicity was associated with overloaded cellular ROS or ferroptosis. However, treatment with antioxidant N-acetylcysteine (NAC) or ferrotatin-1, a ferroptosis inhibitor that can prevent ferroptotic cell death via induction of RSL-3, did not rescue cells from CSG-induced death (Supplementary Fig. 19). Taken together, the current results indicate that CSG kills cells via depleting cellular iron induced by lysosomal dysfunction, regardless of the elevated cellular ROS or ferroptotic pathway, leading to mitochondrial dysfunction and finally cell death.

## Discussion

To elucidate the molecular mechanism associated with the cytotoxicity of the natural product CSG, we conducted label-free target ID using TS-FITGE due to the limited supply and complex structure of CSG. However, target ID using TS-FITGE experiments with CSG were unsuccessful due to a lack of thermal stability shift upon direct engagement of CSG. Previous studies report the thermal stability shift of multi-pass transmembrane proteins upon ligand binding[58–60]; however, in the present study, thermal stability of ATP6V0C was unaltered, even upon the binding of a known ligand (Baf). Therefore, the lack of thermal stability shift precluded the ID of direct target proteins of CSG via TS-FITGE, implying that other thermal stability shift-based label-free target ID methods, such as CETSA[9] (Supplementary Fig. 13) or TPP[10], would also be unlikely to allow ID of CSG target proteins. Though TS-FITGE did not give information about the direct targets of CSG, we cautiously speculate that CSG would bind to and inhibit certain subunits of vacuolar-type ATPases including ATP6V0C that can lead to the inhibition of lysosomal acidification, supported by the inhibition of vacuolar-type

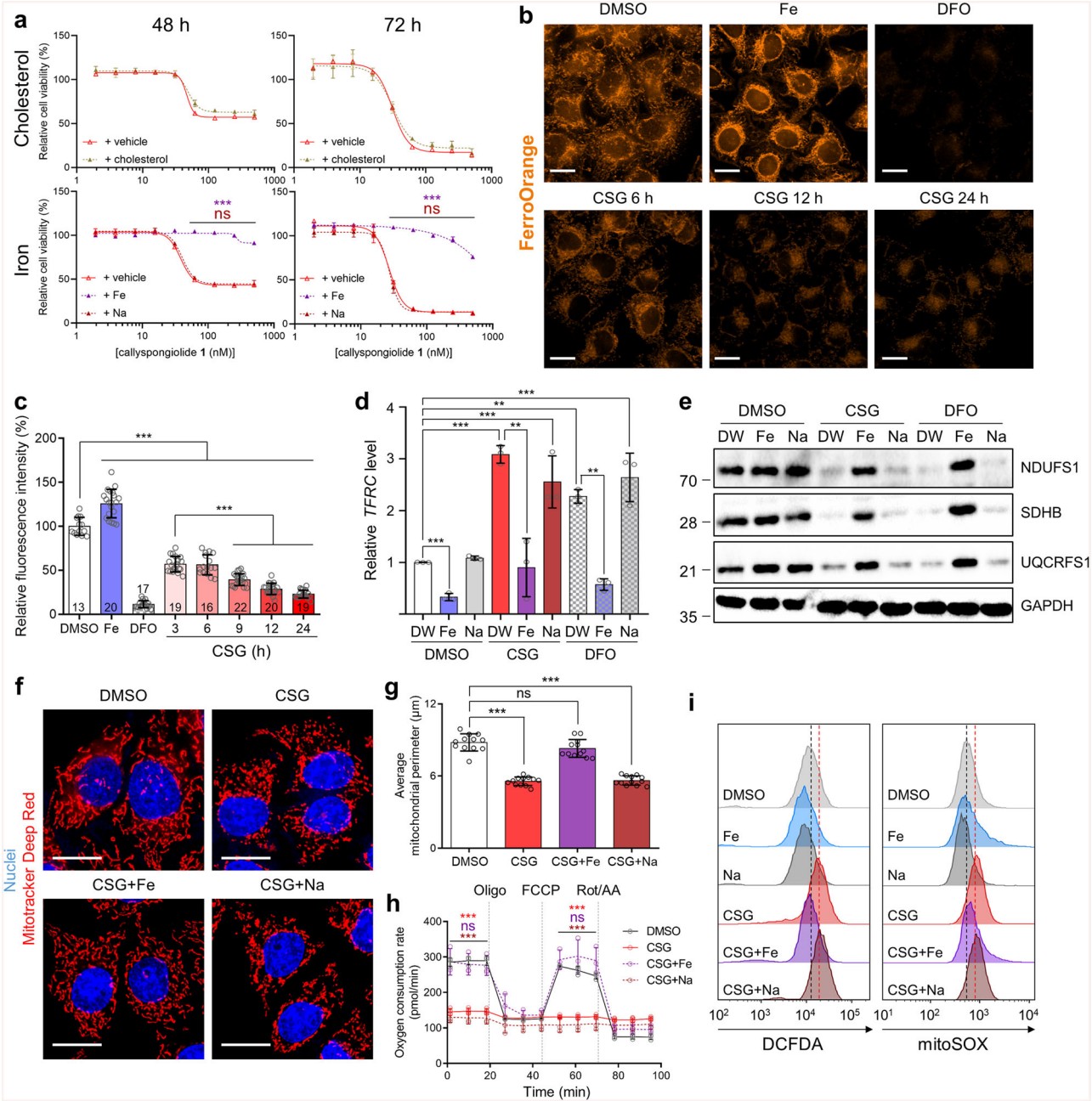

ATPase activity upon CSG treatment, similar to bafilomycin A1 treatment (Fig. 4c), and a previous study reporting vacuolar-type ATPase as a potent target of CSG in yeast[17].

In fact, our TS-FITGE experiment with CSG enabled the unique observation of protein-spot separation on 2-D gels at high temperatures. This phenomenon was only observed with 2-D gel systems, because the protein-spot separation is based on shifts in the isoelectric point of proteins. However, this event is distinct to changes in protein expression or post-translational modification (PTM), because the compound-treatment time used was relatively shorter in duration (≤3 h) than other biological processes, including transcription and translation, and the conditions involved higher-than-normal temperature (>37 °C). Collectively, these observations indicated that this protein-spot separation phenomenon can be only detectable in TS-FITGE, but not in 2-D difference gel electrophoresis (2D-DIGE)[61] or other thermal stability shift-based methods[9,10].

Protein-spot separation was also observed in TS-FITGE using other compounds that induce lysosomal neutralization (Supplementary Fig. 12); however, this phenomenon was neither phenocopied by an iron chelator nor reversed by iron supplementation (Supplementary Fig. 18). In fact, most of the proteins demonstrating spot separation in TS-FITGE were identified as mitochondrial proteins (Supplementary Table 1) and influenced by the altered lysosomal pH, but not by subsequent depletion of cellular iron. To date, lysosomes and mitochondria have been considered distinct organelles, with one functioning as a cellular recycling bin and the other a cellular power plant; however, their physical contact and functional interactions were recently reported as crucial for maintaining their own functions and cellular homeostasis[62–65]. To the best of our knowledge, the present study is the first demonstrating shifts in the isoelectric point of specific proteins at increasing temperature (prior to their melting temperature). Therefore, we speculate that the altered

**Fig. 5 CSG induces cellular iron depletion via lysosome impairment, leading to mitochondrial dysfunction and finally cell death. a** Dose-dependent cell viability following treatment with CSG supplemented with cholesterol (5 μg/mL), ferric citrate (Fe, 200 μM), or sodium citrate (Na, 200 μM). Cell viability is presented as % relative to vehicle-treated cells. Data represent the mean ± SD ($n = 4$). ns: $P > 0.05$, ***$P < 0.001$, one-way ANOVA with Dunnett's *post hoc* test, +vehicle vs. +Fe (purple); +vehicle vs. +Na (brown). **b** Representative live-cell fluorescent images in A549 cells with FerroOrange staining following treatment with supplemental iron (200 μM) or DFO (100 μM) for 24 h or CSG (200 nM) for the indicated times. Scale bar, 10 μm. See also Supplementary Fig. 16. **c** Quantification of FerroOrange fluorescence intensity in (**b**) and Supplementary Fig. 16. Fluorescence intensity is presented as % relative to the DMSO-treated condition. Data represent the mean ± SD (the number of quantified cells is indicated under each bar). ***$P < 0.001$, one-way ANOVA with Bonferroni's *post hoc* test. **d** *TFRC* expression according to qRT-PCR in A549 cells following treatment with CSG (200 nM) or DFO (100 μM) supplemented with or without iron citrate (200 μM) or sodium citrate (200 μM) for 24 h. *TFRC* expression is presented as fold change relative to the vehicle-treated condition. Data represent the mean ± SD ($n = 3$). **$P < 0.01$, ***$P < 0.001$, one-way ANOVA with Dunnett's *post hoc* test. **e** Immunoblotting of NDUFS1, SDHB, and UQCRFS1 in A549 cells following treatment with CSG (200 nM) or DFO (100 μM) and supplementation with or without iron citrate (200 μM) or sodium citrate (200 μM) for 24 h. **f** Representative A549 live-cell fluorescent images of mitochondria using Mitotracker Deep Red following treatment with CSG (200 nM) and supplementation with or without iron citrate (200 μM) or sodium citrate (200 μM). Nuclei were stained with Hoechst 33342. Scale bar, 10 μm. **g** Quantification of the mitochondrial perimeter in (**f**). Data represent the mean ± SD ($n = 12$). ns: $P > 0.05$, ***$P < 0.001$, one-way ANOVA with Dunnett's *post hoc* test. **h** Real-time bioenergetic analysis using the Seahorse XF analyzer of A549 cells following treatment with CSG (200 nM) and supplementation with or without iron citrate (200 μM) or sodium citrate (200 μM) for 24 h. Mitochondrial respiration was measured as oxygen consumption rate (pmol/min). Data represent the mean ± SD ($n = 3$). ns: $P > 0.05$, ***$P < 0.001$, one-way ANOVA with Dunnett's *post hoc* test. Outcomes of DMSO treatment versus CSG treatment are indicated as red, DMSO versus CSG + iron citrate as purple, and DMSO versus CSG + sodium citrate as brown. **i** Fluorescene-activated cell sorting (FACS) analysis of cellular and mitochondrial ROS using DCFDA and MitoSOX in A549 cells following treatment with CSG (200 nM) and supplementation with or without iron citrate (200 μM) or sodium citrate (200 μM). Median outcome of the DMSO-treated condition is indicated by a black-dotted line, and the median outcome of the CSG-treated condition is indicated by a red-dotted line.

lysosomal pH could also promote heat-induced structural modifications and/or changes in PTMs of adjacent proteins, including mitochondrial proteins.

Very recently, Lee and coworkers proposed energy deprivation induced by mitochondrial dysfunction and autophagy-dependent cell death as underlying cytotoxic mechanism of CSG[66]. We also observed similar mitochondrial dysfunction upon CSG treatment (Supplementary Fig. 9), but herein we revealed that cellular iron depletion is a key factor for cytotoxicity caused by CSG. We also observed the autophagy process was partially associated with the cytotoxicity by CSG as shown by ATG5 knockdown study (Supplementary Fig. 14), but the exact mechanism for cytotoxicity by CSG was confirmed as mitochondrial dysfunction followed by cellular iron depletion.

Maintaining iron homeostasis is crucial for cell proliferation and growth, especially, cancer cells show a high demand for iron due to their rapid proliferation[67]. Hence, targeting cellular iron homeostasis, such as the iron deprivation using CSG or DFO, could be a strategy for treating cancer, though systemic toxicity should be considered at the clinical stage. Sandoval-Acuña et al. introduced a mitochondria-targeting functional element to the iron chelator DFO, and named mitoDFO for selective targeting of cancer cells[68]. They showed that mitoDFO killed cells via mitophagy to clear out destabilized mitochondria upon iron depletion, and inhibited tumor growth and metastasis in a mouse model without affecting systemic iron homeostasis. Even though both mitoDFO and CSG killed cells via inducing iron depletion, our data showed that their working mechanisms are different; mitoDFO directly binds and deprives mitochondrial iron leading to mitophagy, but CSG impairs the acidification of lysosome resulting in a hampered autophagy process (Fig. 4b–f), followed by cellular iron depletion. Hence, CSG treatment did not directly induce mitophagy confirmed by unchanged mitochondrial content (Supplementary Fig. 9e), probably due to CSG-mediated lysosomal dysfunction.

As noted, the acidic endosome and lysosome have pivotal roles in supplying bioavailable iron through endocytosis of ferritins and ferritinophagy. Hughes et al. observed that dysfunctional acidic vacuoles in yeast can be a major cause of age-related mitochondrial deterioration through ROS-dependent iron depletion[69]. However, we observed that CSG-mediated cell death

was not rescued by the addition of NAC (Supplementary Fig. 19), and the iron supplementation alleviated CSG-induced ROS generation (Fig. 5i). These results indicated that CSG treatment in mammalian cells resulted in cellular iron deprivation, leading to the impairment of mitochondrial respiration and subsequent ROS generation, rather than ROS generation followed by cellular iron depletion reported in yeast.

In summary, we elucidated the mechanism of action of (−)-CSG, a potent cytotoxic natural compound, through a series of molecular and chemical biology methods, including TS-FITGE. Our results confirmed the applicability of TS-FITGE for monitoring the alteration of cellular environments upon compound treatment on the basis of pattern changes in thermal stability of protein spots, along with its original goal in target identification. Although our 2-D gel-based TS-FITGE method could not suggest protein targets of CSG due to the lack of thermal stability shift in putative target vacuolar-type ATPase, the TS-FITGE results provided the unique phenomenon of protein-spot splitting on the 2-D gels at increasing temperature as an important clue for our mechanistic study. By modifying the TS-FITGE and subsequent mode-of-action studies, we successfully revealed that CSG induces lysosomal and mitochondrial dysfunction as well as autophagy inhibition. Furthermore, we confirmed relationships between impaired lysosomes and cellular iron depletion, given that the mitochondrial dysfunction and subsequent cell death induced by lysosomal dysfunction were rescued upon iron supplementation. These findings demonstrated that CSG kills cells by inducing mitochondrial dysfunction via cellular iron depletion caused by impaired lysosomal acidity independent of known programmed cell death pathways, such as canonical apoptosis, parthanatos, and ferroptosis (Fig. 6). It is worth noting that TS-FITGE can be employed in conjunction with other label-free target ID techniques for direct protein target ID, as well as mechanistic studies of other bioactive compounds.

## Methods

**Materials**. The following media were purchased from Gibco (Gaithersburg, MD, USA): Dulbecco's modified Eagle medium (DMEM; 11995-065), Roswell Park Memorial Institute (RPMI) medium 1640 (11875-093), McCoy's 5A medium (16600-082), advanced DMEM/F12 (12634-010), fetal bovine serum (FBS; 16000-044), and antibiotic–antimycotic (15240-062). The following reagents were purchased from Sigma-Aldrich (St. Louis, MO, USA) and prepared in DMSO (D8418)

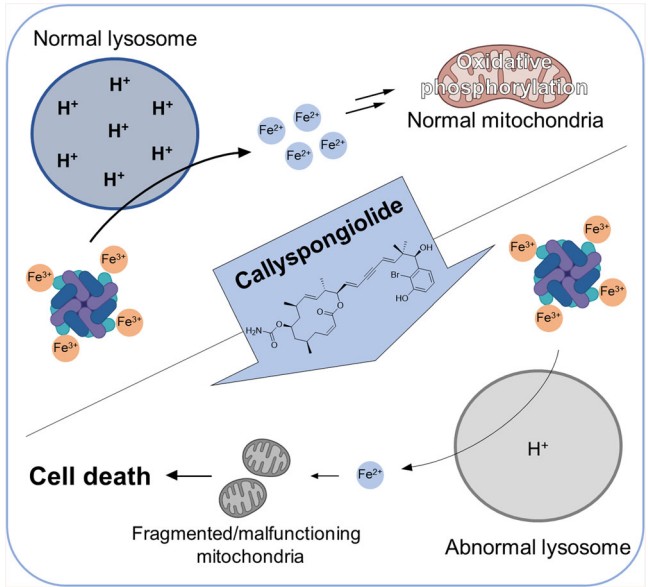

**Fig. 6 CSG kills cells by inducing mitochondrial dysfunction via cellular iron depletion caused by impaired lysosomal acidification.** A cartoon delineating the cytotoxic mechanism of action of the marine natural product CSG.

solution: rotenone (R8875), antimycin A (A8674), oligomycin A (75351), FCCP (C2920), bafilomycin A1 (B1793), deferoxamine (D9533), ferrostatin-1 (SML0583), *1S,3R*-RSL3 (SML2234), and Tween-20 (P9416). Cholesterol was purchased from Sigma-Aldrich (C8667) and prepared in ethanol solution. Ferric citrate was purchased from Sigma-Aldrich (F3388), sodium citrate from Daejung (Seoul, South Korea), and CQ from InvivoGen (tlrl-chq; San Diego, CA, USA) and prepared in distilled water. Phosphate-buffered saline (PBS) was purchased from Wellgene (Seoul, South Korea). IGEPAL CA-630 was purchased from Sigma-Aldrich (I8896). Hoechst 33342 trihydrochloride was purchased from Invitrogen (H3570; Carlsbad, CA, USA). Protease-inhibitor cocktail was purchased from Roche (05056489001; Basel, Switzerland).

**Cell culture**. A549 (human lung cancer cells), HCT116 (human colorectal cancer cells), HEK293T (human embryonic kidney cells), HeLa (human cervical cancer cells), HepG2 (human hepatic cancer cells), Jurkat (human T lymphocyte cells), MCF7 (human breast cancer cells), PC3 (human prostate cancer cells), and SH-SY5Y (human neuroblastoma cells) were purchased from American Type Culture Collection (Manassas, VA, USA). A549, HeLa, Jurkat, MCF7, and PC-3 cells were cultured in RPMI 1640 media. HEK293T and HepG2 cells were cultured in DMEM. HCT116 cells were cultured in McCoy's 5A media. SH-SY5Y cells were cultured in DMEM/F-12K media. All media were supplemented with 10% FBS and 1× antibiotic–antimycotic solution, and all cells were maintained at 5% $CO_2$ in a 37 °C incubator.

**Cell-viability assay**. A day after cell seeding, cells were treated with a range of (−)-CSG or 21-*epi*-(−)-CSG for the indicated times under 5% $CO_2$ at 37 °C. As needed, cholesterol (5 μg/mL), ferric citrate (200 μM), sodium citrate (200 μM), or ferrostatin-1 (1 μM) was applied as a co-treatment. To induce ferroptosis, *1S,3R*-RSL3 (1 μM) was used to treat A549 cells in the absence or presence of ferrostatin-1 (1 or 5 μM) for 24 h. Cell viability was assessed by WST assay with EZ-Cytox (DoGenBio, Seoul, South Korea). Relative cell viability was normalized to the vehicle-treated condition.

**Immunoblot**. Cells were collected and lysed in RIPA buffer [50 mM Tris-HCl (pH 7.8), 150 mM NaCl, 0.5% sodium deoxycholate, 1% IGEPAL CA-630, and protease-inhibitor cocktail] at 4 °C for 20 min. The soluble fraction was collected after centrifugation at 20,000 *g* for 20 min at 4 °C, and total protein concentration was determined using a Pierce BCA protein assay kit (Thermo Fisher Scientific, Waltham, MA, USA). Equal amounts of each lysate were separated by SDS-PAGE and transferred to PVDF membranes (Bio-Rad Laboratories, Hercules, CA, USA). After blocking with 2% bovine serum albumin (BSA; MP Biomedicals, Irvine, CA, USA) in Tris-buffered saline containing 1% Tween-20 (TBST) for 1 h, the membranes were incubated with primary antibodies in 1% BSA in TBST overnight at 4 °C. The next day, membranes were washed three times with TBST and incubated with anti-rabbit IgG horseradish peroxidase (HRP)-linked (#7074; Cell Signaling Technology, Danvers, MD, USA) or anti-mouse IgG HRP-linked (#7076; Cell Signaling Technology) in 1% BSA and TBST at room temperature for 1 h. After five

washes with TBST, protein bands were developed with Prime enhanced chemiluminescence western blot detection reagent (Cytiva, Marlborough, MA, USA) and detected using a ChemiDoc system (Bio-Rad Laboratories). The following antibodies were purchased and used as primary antibodies for immunoblotting: anti-caspase-3 (#9662; Cell Signaling Technology), anti-caspase-9 (#9502; Cell Signaling Technology), anti-PARP (#9532; Cell Signaling Technology), anti-SDHA (#11998; Cell Signaling Technology), anti-AIF (sc-13116; Santa Cruz Biotechnology, Dallas, TX, USA), anti-ATP5F1A (sc-136178; Santa Cruz Biotechnology), anti-UQCRC2 (sc-390378; Santa Cruz Biotechnology), anti-TOMM40 (PA5-57575; Invitrogen), anti-PP2AC (#2038; Cell Signaling Technology), anti-ATP5F1 (sc-514419; Santa Cruz Biotechnology), anti-NDUFB10 (PA5-51179; Invitrogen), anti-BTF3 (PA5-63299; Invitrogen), anti-ATP6V0C (PA5-23972; Invitrogen), anti-ATG5 (ab108327; Abcam, Cambridge, UK), anti-LC3B (#83506; Cell Signaling Technology), anti-p62 (ab91526; Abcam), anti-NDUFS1 (sc-271510; Santa Cruz Biotechnology), anti-SDHB (sc-271548; Santa Cruz Biotechnology), anti-UQCRFS1 (sc-271609; Santa Cruz Biotechnology), anti-FTH1 (#3998, Cell Signaling Technology), anti-TfR1 (#13113, Cell Signaling Technology), and anti-glyceraldehyde 3-phosphate dehydrogenase (#2118; Cell Signaling Technology).

**Flow cytometry with annexin V-PI staining**. A day after A549 cells were seeded on 12-well plate (Corning), the cells were treated with CSG (200 nM) or etoposide (50 μM) (E1383; Sigma-Aldrich) for indicated times under 5% $CO_2$ at 37 °C. The cells were trypsinized, and equal number of the cells in each condition was subjected to Annexin V-FITC apoptosis detection kit (ab14085; Abcam), according to manufacturer instructions. Briefly, collected cells were suspended in the Binding buffer, then incubated with Annexin V-FITC antibody and propidium iodide at room temperature for 5 min in dark. Fluorescently labeled cells were analyzed by FACS Aria II (BD Bioscience, San Jose, CA, USA) installed at the National Center for Inter-university Research Facilities (NCIRF) at Seoul National University, Seoul, Korea. Data were analyzed using FlowJo X 10.0.7r2 software (FlowJo, LLC, Ashland, OR, USA).

**Immunofluorescence of AIF and Line profile**. A day after A549 cells were seeded on chambered coverglass (Lab-tek), CSG (200 nM) was treated for indicated times under 5% $CO_2$ at 37 °C. After washed with PBS once, the cells were fixed with 4% paraformaldehyde (Sigma-Aldrich) prepared in PBS solution for 15 min at room temperature. The fixation solution was discarded, then the cells were washed with PBS twice. For cell permeabilization, 0.1% Triton X-100 (Sigma-Aldrich) prepared in PBS solution was treated to the cells for 15 min at room temperature. The permeabilization solution was discarded, then the cells were washed with PBS twice. After blocking with 4% BSA solution in PBS, the cells were incubated with primary antibodies (1:300 in 1% BSA in PBS)——anti-AIF and anti-TOMM20 (ab186735; Abcam)——overnight at 4 °C. The next day, the cells were washed with PBS 3 times, then incubated with secondary antibodies (1:400 in 1% BSA in PBS)——FITC-conjugated anti-mouse (ab6785; Abcam) and TRITC-conjugated anti-rabbit (ab6718, Abcam)——for 1 h at room temperature in dark. After the cells were washed with PBS 5 times, nuclei were stained with Hoechst 33342 (1:5000 in PBS) for 15 min, and the staining solution was washed out. Fluorescent microscopy imaging was performed with DeltaVision Elite imaging system (Cytiva). The system was equipped with 60× objective lenses of Olympus IX-71 inverted microscope with PLAN APO 60×/Oil (PLAPON60×O), 1.42 NA, WD 0.15 mm. sCMOS camera and InSightSSI fluorescence illumination module. Fluorescent filter set were as followed: DAPI/DAPI (excitation: 390/18 nm, emission: 435/48 nm), FITC/FITC (excitation: 475/28 nm, emission: 525/48 nm), and TRITC/TRITC (excitation: 542/27 nm, emission: 594/45 nm). Images were analyzed with softWoRx software (Cytiva), and Color profile was performed with ImageJ software by National Institute of Health.

**TUNEL assay**. A day after cells were seeded on chambered coverglass, CSG (200 nM) was treated for indicated times under 5% $CO_2$ at 37 °C. DNA fragmentation was observed with in situ direct DNA fragmentation (TUNEL) assay kit (ab66108; Abcam) according to manufacturer instructions. Briefly, after the media was aspirated, the cells were washed with PBS. For the fixation, 1% paraformaldehyde solution in PBS was added, and incubated for 15 min at 4 °C. After washed with PBS once, 70% ethanol solution in distilled water (DW) was added to the cells, then the cells were incubated for 30 min at 4 °C. As a positive control, the cells were treated with DNase I (Thermo Scientific) for 30 min at room temperature. All cells were washed with the Wash buffer twice. DNA labeling solution was treated for 1 h at 37 °C. The Rinse buffer was added to the labeling solution, and the cells were washed with rinse buffer again. Nuclei was stained with Hoechst 33342 (1:5000 in DW) supplemented with RNase A for 30 min at room temperature in dark. After washed with PBS twice, fluorescent microscopy imaging was performed with DeltaVision imaging system as mentioned above using DAPI/DAPI and FITC/FITC filter sets. Images were analyzed with softWoRx software and ImageJ software.

**TS-FITGE**. The general procedure for TS-FITGE was followed as previously described[11]. For live-cell TS-FITGE, cells were trypsinized and suspended in serum-free RPMI (containing only antibiotic–antimycotic) in 50-mL conical tubes (Corning, Corning, NY, USA). As required, Rot (10 μM), AA (10 μM), Oligo

(20 μM), or FCCP (20 μM) was pretreated for 20 min. DMSO (0.1%), CSG (200 nM), FCCP (20 μM), Baf (100 nM), CQ (100 μM), or DFO (100 μM) were added to the cells, as required, and ferric citrate (200 μM) was added to both groups, followed by a 3-h incubation under 5% $CO_2$ at 37 °C. The cells were then heat shocked at the indicated temperatures for 3 min and then cooled at 25 °C for 3 min. After washing with PBS, the cells were lysed by the freeze–thaw method in liquid nitrogen three times in lysis buffer (0.4% IGEPAL CA-630 in PBS supplemented with a protease-inhibitor cocktail). Soluble proteins were isolated from cell debris by centrifugation at $20,000 \times g$ for 20 min at 4 °C and quantified by BCA assay. For lysate TS-FITGE, A549 cells were collected, lysed, and quantified, and 2 mg/mL of cell lysate was treated with DMSO (0.1%) or CSG (200 nM) for 2 h at room temperature. The lysate was then aliquoted and heat shocked, followed by isolation of the soluble fraction and processing as described. Protein (50 μg) was precipitated in cold acetone (−20 °C), followed by centrifugation at $20,000 \times g$ for 7 min at 4 °C. The residual pellet was then suspended in 10 μL of labeling buffer [30 mM Tris-HCl (pH 8.6), 2 M thiourea, 7 M urea, and 4% CHAPS], and 1 μL of 0.4 mM Cy3-NHS or Cy5-NHS was added to the protein solutions and incubated at 4 °C for 45 min. The dye-conjugated proteins were then precipitated in cold acetone (−20 °C), followed by centrifugation at $20,000 \times g$ for 7 min at 4 °C and pellet resuspension in rehydration buffer (7 M urea, 2 M thiourea, 2% CHAPS, 40 mM DTT, and 1% IPG buffer). The Cy3- and Cy5-labeled proteins at the same temperature were then mixed and loaded on a 24-cm Immobiline Drystip gel (Cytiva), and isoelectric focusing was conducted by Ettan IPGphor 3 (Cytiva) according to manufacturer instructions. After isoelectric focusing, proteins in the strip gel were resolved using the Ettan DALTsix electrophoresis system (Cytiva), and fluorescence on the gels was scanned with a Sapphire Biomolecular Imager (Azure Biosystems, Dublin, CA, USA). The protein-spot location and fluorescence intensity were analyzed by Melanie software (v.9.2.3; Cytiva). The Cy5:Cy3 intensity ratio was normalized, and data were presented as box-and-whisker plots using GraphPad Prism 8 software (GraphPad software, San Diego, CA, USA).

**In-gel digestion and mass spectrometry.** The protein spots from silver-stained gel were excised, destained, and digested with trypsin. The mixture was evaporated in SpeedVac and then dissolved in 10% acetonitrile with 0.1% formic acid. The resulting peptides were desalted in a trap column (180 μm × 20 mm, Symmetry C18) and separated on a C18 reversed-phase analytical column (75 μm × 200 mm, 1.7 μm, BEH130 C18) (Waters) with an electrospray ionization Pico Tip (±10 μm i.d.) (New objective). The data were converted to .pkl files by Protein Lynx Gobal Server and searched by MASCOT engine with the SwissProt database.

**CETSA.** A549 cells were trypsinized, collected, and suspended in serum-free RPMI in 50-mL conical tubes. The cells were treated with DMSO (0.1%), CSG (200 nM), or baf (100 nM) for 3 h under 5% $CO_2$ at 37 °C. The cell suspension was aliquoted, heat shocked at a range of temperature for 3 min, then cooled at 25 °C for 3 min. After washed with PBS, the heated cells were lysed by freeze&thaw in liquid nitrogen 3 times in lysis buffer. The same volume of soluble proteome isolated by centrifugation at $20,000 \times g$ for 20 min at 4 °C was combined with SDS buffer, then subjected to immunoblotting.

**Gene knockdown.** Short interfering RNA (siRNA) oligonucleotides against *PP2AC* (s10957; Ambion) and *ATG5* (s18158; Ambion) were transfected to A549 cells for 2 days under 5% $CO_2$ at 37 °C using Lipofectamine RNAiMAX (Invitrogen) and Opti-MEM (Gibco) according to manufacturer instruction. Scramble RNA (Bioneer, South Korea) was used as a negative control. Gene knockdown was confirmed by immunoblotting. For the experiments to assess the activity change of CSG after gene knockdown, cells were transfected with siRNA for 2 days and then treated with a range of CSG for indicated times, followed by cell-viability assay.

**Assessment of mitochondrial morphology and perimeter quantification.** A day after A549 cells were seeded on chambered coverglass, CSG (200 nM) was applied for the indicated times with incubation under 5% $CO_2$ at 37 °C. As needed, ferric citrate (200 μM) or sodium citrate (200 μM) was applied together. The media were then aspirated, the cells were washed with PBS, and nuclei and mitochondria were stained using Hoechst 33342 (1:5000) and MitoTracker Deep Red (200 nM; M22426; Invitrogen], respectively, in culture media for 20 min. The staining media were then aspirated, and the cells were washed with culture media twice. Live-cell fluorescence microscopy imaging was performed with the DeltaVision Elite imaging system (Cytiva). The system was equipped with 60× objective lens of Olympus IX-71 inverted microscope with PLAN APO ×60/Oil (PLAPON60×O), 1.42 NA, WD 0.15 mm. sCMOS camera and InSightSSI fluorescence illumination module. Fluorescent filter set were as followed: 4′,6-diamidino-2-phenylindole (DAPI)/DAPI (excitation: 390/18 nm; emission: 435/48 nm) and Cy5/Cy5 (excitation: 632/22 nm; emission: 676/34 nm) installed with a $CO_2$-supporting chamber and an objective air heater at 37 °C for live-cell imaging. Images were analyzed with softWoRx software. For quantification of the mitochondrial perimeter, images were analyzed with ImageJ software (National Institutes of Health, Bethesda, MD, USA) using a previously described protocol, with minor modifications[70]. Briefly, background images were obtained from *Median* filter (Process > Filters > median…) using Radius 7, and the original images were subtracted from the background

images to correct for inhomogeneous background. The mitochondrial signal of the resulting images was adjusted according to an intensity threshold (Image > adjust > threshold…) using the *Triangle* algorithm, and the perimeter of mitochondria was quantified by *Analyze particles* (Analyze > analyze particles…) using a size filter of 10-infinity (pixel).

**Bioenergetic analysis.** The OCR was measured using a Seahorse XFe24 extracellular flux analyzer (Seahorse Bioscience, Billerica, MA, USA) according to manufacturer instructions. Briefly, $1 \times 10^4$ A549 cells were seeded on XFe24 cell culture microplates and cultured for 1 day under 5% $CO_2$ at 37 °C. CSG (200 nM) was added to the cells in the presence or absence of ferric citrate (200 μM) or sodium citrate (200 μM) and incubated for the indicated times. To measure mitochondrial respiration, Oligo (0.5 μM), FCCP (1 μM), and Rot/AA (0.5 μM) were added sequentially through the injection ports while measuring OCR.

**Mitochondrial DNA quantification.** A day after A549 cells were seeded on 6-well plate (Corning), CSG (200 nM) was treated for indicated times under 5% $CO_2$ at 37 °C. The media was aspirated, then the cells were washed with PBS. Cells were collected, and genomic DNA was prepared with DNeasy DNA extraction kit (Qiagen) according to the manufacturer's instructions. The DNA was quantified using NanoVue (Cytiva). 20 ng of genomic DNA was subjected to quantitative PCR (qPCR), combined with 1× KAPA SYBR FAST ABI Prism qPCR master mix (KAPA Biosystems), and forward/reverse primers (200 nM) against human mitochondrial DNA (forward: 5′-CCCCACAAACCCCATTACTAAACCC A-3′; reverse: 5′-TTTCATCATGCGGAGATGTTGGATGG-3′) or human β-actin (forward: 5′-TGTGTGGGGAGCTGTCACAT-3′; reverse: 5′-CGCCTAGAAGCATTTGCGGT-3′) for a final volume 20 μL with nuclease-free water. The PCR was conducted on StepOnePlus (Applied Biosystems), and the PCR cycling was used; initial denaturing at 95 °C for 3 min, followed by 40 cycles of denaturing at 95 °C for 3 s, and extension at 60 °C for 25 s. The data were analyzed by the comparative $C_t$ method, and the mitochondrial DNA levels were normalized to *β-actin* levels.

**Assessment of mitochondrial membrane potential and lysosome acidity.** A day after A549 cells were seeded on chambered coverglass, FCCP (20 μM), Baf (20 nM), or CSG (200 nM) was added to the cells for incubation for 24 h under 5% $CO_2$ at 37 °C. The media were then aspirated, and the cells were washed with fresh culture media. Nuclei, mitochondrial membrane potential, and lysosome membrane potential were stained using Hoechst 33342 (1:5000), TMRE (500 nM; T669; Invitrogen), and Lysotracker Red DND-99 (L7528; Invitrogen), respectively, in the culture media for 30 min under 5% $CO_2$ at 37 °C. The media were then aspirated, and the cells were washed with culture media twice. Live-cell fluorescence microscopy imaging was performed with the DeltaVision imaging system (Cytiva) in the presence of FCCP, Baf, or CSG using DAPI/DAPI, tetramethylrhodamine (TRITC)/TRITC (excitation: 542/27 nm; emission: 594/45 nm), and A594/A594 (excitation: 575/25 nm; emission: 632/60 nm) filter sets installed with a $CO_2$-supporting chamber and an objective air heater at 37 °C for live-cell imaging. Images were analyzed with softWoRx software.

**In vitro vacuolar-type ATPase activity assay.** The experiment was performed as previously described[30]. Briefly, confluent HEK293T cells on two 150-pi dishes were incubated for 16 h with 25 μg/mL 70 kDa Dextran conjugated to Oregon Green 514 (Dx-OG514, D7176; Invitrogen). The cells were washed with PBS twice, and incubated in serum-free DMEM to allow accumulation in lysosome of Dx-OG514. After addition of 1 μM of FCCP for 15 min for dissipating proton gradient across lysosomal membrane, cells were then collected and lysed by syringing through 23 G needle in fractionation buffer (140 mM KCl, 1 mM EGTA, 20 mM HEPES, 50 mM sucrose, 5 mM glucose, 1 μM FCCP, pH 7.4, supplemented with protease-inhibitor cocktail). Lysed cells were centrifuged at 1,700 r.p.m. for 10 min at 4 °C, then the resulting supernatant was centrifuged at $20,000 \times g$ for 20 min at 4 °C, yielding a pellet containing organellar fraction with lysosomal membrane. The pellet was resuspended in fractionation buffer devoid of FCCP, and transferred to black 96-well plate (Corning). Baf (20 nM) or CSG (200 nM) were added to each well, followed by addition of Na$_2$ATP (5 mM) and MgCl$_2$ (5 mM). The fluorescence was measured by Synergy HTX microplate reader (BioTek) with following filter set; excitation: 485/20 nm, emission: 528/20 nm.

**mCherry-GFP-LC3B imaging.** A day after A549 cells were seeded on chambered coverglass, the mCherry-GFP-LC3B plasmid (pBabe vector) was transfected into the cells for 6 h under 5% $CO_2$ at 37 °C using Lipofectamine LTX plus (Invitrogen) and Opti-MEM (Gibco) according to manufacturer instructions. After the media were aspirated, the cells were washed with PBS. Culture media containing Rapa (500 nM), Baf (20 nM), or CSG (200 nM) were then added to the cells and incubated for 24 h under 5% $CO_2$ at 37 °C. Following media aspiration, nuclei were stained with Hoechst 33342 (1:5000) in the culture media. After the cells were washed with culture media twice, they were sequentially subjected to live-cell fluorescence microscopy imaging with the DeltaVision imaging system (Cytiva) using DAPI/DAPI, fluorescein isothiocyanate (FITC)/FITC (excitation: 475/28 nm; emission: 523/36 nm), and A594/A594 filter sets installed with a $CO_2$-supporting

chamber and an objective air heater at 37 °C for live-cell imaging. Images were analyzed with softWoRx software.

**FerroOrange staining**. A day after A549 cells were seeded on chambered coverglass, ferric citrate (200 μM) or DFO (100 μM) were added for 24 h, and CSG (200 nM) was added for incubation for the indicated times under 5% $CO_2$ at 37 °C. FerroOrange (F374; Dojindo, Kumamoto, Japan) staining was conducted according to manufacturer instructions. Briefly, the media were aspirated, then the cells were washed with serum-free RPMI media three times. Cellular iron was stained with FerroOrange (1 μM) in serum-free media for 30 min under 5% $CO_2$ at 37 °C. Without a washing step, the cells were then directly subjected to live-cell fluorescence microscopy imaging using the DeltaVision imaging system (Cytiva) using TRITC/TRITC filter sets installed with a $CO_2$-supporting chamber and an objective air heater at 37 °C for live-cell imaging. Images were analyzed with softWoRx software. For quantification of fluorescence intensity, the border of each cell was extrapolated using bright-field images, and the FerroOrange signal in each cell was quantified with ImageJ software.

**Heme content assay**. Cellular heme level was measured by fluorometric method as previously published[71,72] with minor modifications. A day after A549 cells were seeded on 6-well plate (Corning), CSG (200 nM) or DFO (100 μM) was treated in the absence or presence of ferric citrate (200 μM) or sodium citrate (200 μM) for 24 h under 5% $CO_2$ at 37 °C. The media was aspirated, then the cells were washed with PBS. Cells were collected, then lysed in Triton X-100 lysis buffer (50 mM Tris-HCl pH 7.6, 100 mM NaCl, 1% Triton X-100, and protease-inhibitor cocktail). The soluble fraction was collected after centrifugation at $20,000 \times g$ for 20 min at 4 °C, and total protein concentration was determined using Pierce BCA protein assay kit. 1 volume of cell lysate was combined with 20 volume of supersaturated oxalic acid (about 2 M), then the samples were immediately boiled at 95 °C for 30 min. A parallel set of samples were left unheated for blank. The samples were cooled and transferred to black 96-well plate, then the fluorescence from protoporphyrin IX was measured under excitation at 400 nm and emission at 620(10) nm using Tecan Spark multimode microplate reader, then normalized to protein concentration.

**RNA extraction and quantitative reverse transcription PCR (qRT-PCR)**. A day after A549 cells were seeded on 12-well plates (Corning), and CSG (200 nM) or DFO (100 μM) was added in the presence or absence of sodium citrate (200 μM) or ferric citrate (200 μM) for incubation for 24 h under 5% $CO_2$ at 37 °C. The media were then aspirated, and the cells were washed with PBS. Subsequently, total RNA was extracted using an RNeasy kit (Qiagen, Hilden, Germany) according to manufacturer instructions, RNA was quantified using a NanoVue spectrophotometer (Biochrom, Cambridge, UK), and 1 μg of RNA was used for synthesis of cDNA in a final volume of 20 μL (with nuclease-free water) using AccuPower CycleScript RT PreMix dT20 (Bioneer, Daejeon, South Korea) according to manufacturer instructions. cDNA (1 μg) was then subjected to qRT-PCR in a solution containing 1× KAPA SYBR FAST ABI Prism qPCR master mix (KAPA Biosystems, Wilmington, MA, USA) and forward/reverse primers (200 nM) targeting human *TFRC* (forward, 5′-ATCGGTTGGTGCCACT-GAATGG-3′ and reverse, 5′-ACAACAGTGGGCTGGCAGAAAC-3′) or human *β-actin* at a final volume 20 μL (with nuclease-free water). PCR was conducted on a StepOnePlus system (Applied Biosystems, Foster City, CA, USA) according to the following cycling conditions: initial denaturation at 95 °C for 3 min, followed by 40 cycles at 95 °C for 3 s and extension at 60 °C for 25 s. Data were analyzed by the comparative $C_t$ method and normalized to *β-actin* level.

**Flow cytometric analysis with DCFDA and mitoSOX**. A day after A549 cells were seeded on 6-well plates, the cells were treated with ferric citrate (200 μM) or sodium citrate (200 μM) in the presence or absence of CSG (200 nM) for 24 h under 5% $CO_2$ at 37 °C. The media were then aspirated, and the cells were washed with PBS, collected, suspended in PBS containing 2% FBS, and stained with DCFDA (10 μM; D6883; Sigma-Aldrich) or MitoSOX (2.5 μM; M36008; Invitrogen) for 30 min under 5% $CO_2$ at 37 °C. The resulting cells were subjected to flow cytometric analysis using a FACS Aria II system (BD Biosciences, Franklin Lakes, NJ, USA) installed at the National Center for Inter-university Research Facilities at Seoul National University. Data were analyzed using FlowJo X 10.0.7r2 software (FlowJo, LLC, Ashland, OR, USA).

**Statistics and reproducibility**. All experimental data are presented as the mean ± standard deviation (SD) of three or more biologically independent experiments. The number of replicates is included in the corresponding figure legends. Statistical analyses were performed using GraphPad Prism 8 software (GraphPad Software). Statistical significance was assessed using one-way analysis of variance (ANOVA) with Dunnett's *post hoc* analysis or Bonferroni's *post hoc* analysis. A $P < 0.05$ was considered significant.

**Reporting summary**. Further information on research design is available in the Nature Research Reporting Summary linked to this article.

## Data availability
All data generated and analyzed during this study are included in this article. The uncropped immunoblots underlying Figs. 4d, e, 5e, Supplementary Figs. 2a, 6, 7c, 13, 14c, and 17a are provided in Supplementary Information (Supplementary Figs. 20–28), and the source data underlying Table 1, Figs. 4c, 5a, c, d, g, h, Supplementary Figs 1, 2c, 2d, 7b, 9c, 9d, 9e, 14b, 15, 17b, 19a, 19b, and 19c are provided in Supplementary Data 1. Additional details are also available from the corresponding authors upon reasonable request.

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

## Acknowledgements

This work was supported by the National Creative Research Initiative Grant (2014R1A3A2030423) and the Bio & Medical Technology Development Program (2012M3A9C4048780) through the National Research Foundation of Korea (NRF) funded by the Korean Government (Ministry of Science & ICT). J.H. is grateful for the Fostering Core Leaders of the Future Basic Science Program/Global Ph.D. Fellowship Program (NRF-2016H1A2A1907084). We thank Prof. Tao Ye (Peking University) for kindly providing (−)-CSG and 21-*epi*-(−)-CSG. Figure 6 was created with Biorender.com.

## Author contributions

J.H. and S.B.P. conceived and designed the project. J.H. performed all experiments and analyzed all data. J.H. and S.B.P. wrote the manuscript.

## Competing interests

The authors declare no competing interests.
