## [Transparent Peer Review File · Communications Biology]

Reviewers' comments:

Reviewer #1 (Remarks to the Author):

I have read the manuscript entitled "Callyspongiolide Kills Cells by Inducing Mitochondrial Catastrophe via Cellular Iron Depletion" with interest and although I believe that the core findings are strong, I would have several comments and suggestions:

1. I would modify the title a bit so that it reflects that the core finding is actually that the compound affects lysosomal acidification which then leads to dysfunctional release of the iron from lysosome, leading to mitochondria dysfunction and eventually cell death. Also, since the authors show that FCCP and bafilomycin treatment induces similar changes as callyspongiolide (CSG) but an iron chelator DFO does not elicit these changes, it shows that the observed shifts in TS-FITGE, it is not clear whether iron chelation actually leads to cell death similarly to CSG. Thus I would include DFO treatment in those experiment where authors show that iron supplementation protects cells from cell death. Also, I would rather call the observed effect on mitochondria mitochondrial dysfunction rather than „mitochondrial catastrophe“.

2. I believe it would be beneficial to show that the cells are functionally iron deficient, e.g. measure the amount of key proteins that are affected by the IRP/IRE system – transferrin receptor 1 and ferritin. I would thus suggest to perform the western blot analysis of TFR1 and FTH and show that the levels of FTH decreased while TFRC is increased. TFRC data on mRNA level are shown but protein is missing.

3. Mitochondrial fragmentation is mentioned and shown on the Figure 4a but is not quantified. I believe that using ImageJ it is possible to quantify the number of mitochondrial objects and their size and based on that the authors could obtain the quantifiable measure of mitochondrial fragmentation.

4. If the mitochondrial function is compromised and iron supplementation restores the phenotype- Is the FeS cluster and heme biosynthesis in mitochondria affected? Could authors measure heme levels by a fluorescence based method (DOI: 10.1021/ac60228a014)?

5. Recently, an article that described specific cell death induction by mitochondrially targeted DFO (mitoDFO) has been published (doi: 10.1158/0008-5472.CAN-20-1628). Although it induces similar output, e.g. mitochondrial dysfunction and cell death induction by iron depletion, the mechanism is apparently different, as CSG main effect seem to be the block of lysosomal acidification and also inhibition of autophagy. On the other hand the mitoDFO compound rather induces mitophagy and decreases the level of p62 and apparently lysosomal activity is retained. Could the authors comment on that and discuss it? Would they be able to assess mitophagy with the Mitophagy dye from Dojindo to define whether in CSG effect it plays some role?

6. Authors provide a Seahorse analysis with OCR reduction after CSG treatment, they also show higher ROS levels after CSG, Could they assess the assembly and structure of mitochondrial respiratory supercomplexes by blue native electrophoresis (BNE)??

7. In general, the paper is very interesting but the question is whether there are data that would suggest that the observed effect is specific to cancer cells and does not affect for example non-malignant normal human cells such as fibroblasts? Are there any data on its in vivo effect and toxicity? I have not found such studies but the authors could correct me and should discuss this as well.

Reviewer #2 (Remarks to the Author):

This manuscript reveals that the cytotoxicity of callyspongiolide (CSG) on cancer cells is dependent on the induction of mitochondrial catastrophe through cellular iron depletion caused by lysosomal deacidification. Of note, TS-FITGE, a label-free target identification method, was employed in discovering ten possible direct protein targets of CSG. The manuscript is clearly written, but some revisions are required to improve the overall quality:

1) Previous study reports that epi-CSG is more potent than the natural form of CSG in killing cancer

cells. The authors may explain why CSG is more potent in this study.

- 2) "CSG-mediated cytotoxicity was unaffected by PP2AC depletion" is not accurate, because observable expression of PP2AC still appeared after siRNA treatment as the western blots show. The authors may knockdown other 9 proteins identified by TS-FITGE to see whether it affects CSG activity.
- 3) "... protein spot separation was not relevant to the function of the electron-transport chain". The conclusion is contradictory to the fact that 10 protein spots were identified by TS-FITGE, belonging to the mitochondrial electron-transport system.
- 4) "This observation implied that FCCP treatment phenocopied the TS-FITGE outcomes for CSG". Are these proteins the same as that identified from CSG treated cells? The authors may employ tandem mass spectroscopy to characterize them.
- 6) "CSG was slightly less toxic under ATG5 depletion". Western blots result indicates that there is still some expression of ATG5 after siRNA treatment.
- 7) In Fig. 5 b-c, the decreasing immunofluorescence maybe caused by the cytotoxicity of staining. The authors should use better controls.
- 8) It would be helpful if the authors predict possible target(s) of CSG in killing cancer cells.
- 9) In the discussion part: "The results confirmed the applicability of TS-FITGE for the ID of direct target proteins" is not accurate, as the authors did not find direct targets of CSG using TS-FITGE in this manuscript.

Reviewer #3 (Remarks to the Author):

1. The current study offered new insights for the mode of action of the natural product callyspongiolide (CSG). The authors confirmed the antitumor activities of CSG and tried to identify the target protein. Moreover, they found that CSG is a potent inhibitor of the Vacuolar ATPase in mammalian cells, which is consistent with findings in yeast (ref. 17). In addition, they found iron depletion by CSG leads to cell death. These important novel findings are surely interesting to readers of this journal and draw more discussions in the field.
2. This reviewer is not familiar with methods for label-free target identification, especially TS-FITGE (thermal stability shift-based fluorescence difference in two-dimensional (2-D) gel electrophoresis). No further comments on those experiments involving TS-FITGE should be given. But it's clear that this method has been mentioned quite often in the manuscript and used initially to look for protein target of CSG and eventually led the authors "questioned whether CSG affects mitochondrial morphology and respiratory function". Maybe it's appropriate they add TS-FITGE to the title.
3. The current manuscript described experiments supporting CSG as an inhibitor of Vacuolar ATPase in HEK293T cells. This confirms that CSG is a potent inhibitor of the Vacuolar ATPase (in yeast, ref. 17). Then, abnormal lysosome acidity seems to be the natural result of inhibiting Vacuolar ATPase. But the authors went to TS-FITGE data for clues. That looks like an unnecessary detour.
4. The authors concluded that CSG kills cells by inducing mitochondrial catastrophe via cellular iron depletion, based on FerroOrange imaging and ferric citrate supplementation experiments. As compared to the view of energy deprivation induced by mitochondrial dysfunction(ref. 62), the current study offered a different mechanism with new evidences, although no further discussions about alternatives were found in the manuscript. I am thinking of the report by Hughes et al. (Cell, 2020, p296-310)-they pointed out" cysteine depletion or iron supplementation restores mitochondrial health in vacuole-impaired cells and prevents mitochondrial decline during aging".

Response to Reviewer's comment

We are much appreciating the rapid review of our paper. We are also grateful for positive, constructive comments and suggestions. We address the reviewers' comments below in blue, and present the revised parts in green.

Reviewer #1's comment

I have read the manuscript entitled "Callyspongiolide Kills Cells by Inducing Mitochondrial Catastrophe via Cellular Iron Depletion" with interest and although I believe that the core findings are strong, I would have several comments and suggestions:

1. I would modify the title a bit so that it reflects that the core finding is actually that the compound affects lysosomal acidification which then leads to dysfunctional release of the iron from lysosome, leading to mitochondria dysfunction and eventually cell death. Also, since the authors show that FCCP and bafilomycin treatment induces similar changes as callyspongiolide (CSG) but an iron chelator DFO does not elicit these changes, it shows that the observed shifts in TS-FITGE, it is not clear whether iron chelation actually leads to cell death similarly to CSG. Thus I would include DFO treatment in those experiment where authors show that iron supplementation protects cells from cell death. Also, I would rather call the observed effect on mitochondria mitochondrial dysfunction rather than "mitochondrial catastrophe".

Our response

We appreciate your thoughtful suggestions. Based on your comment, we added the cell viability result upon DFO treatment in A549 cells (revised Supplementary Fig. 15) to emphasize that CSG-mediate cell death is resulted from cellular iron depletion like DFO treatment. Similar to CSG treatment, DFO treatment showed gradual cytotoxicity over time (no significant cytotoxicity for 24-h treatment, but time-dependent increase in cytotoxicity for 48- to 72-h treatment), which was rescued by ferric citrate supplementation, but not by sodium citrate.

Changes in main text

(page 11, line 263)

"Indeed, iron supplementation with ferric citrate, but not cholesterol or sodium citrate supplementation, prevented cell death caused by CSG (Fig. 5a). Similarly, cellular iron deprivation by the iron chelator deferoxamine (DFO) also showed gradual cytotoxicity over time, and the cells were rescued by ferric citrate supplementation, not by sodium citrate (Supplementary Fig. 15). These results indicate that cellular iron depletion underlies CSG-mediated cytotoxicity."

Supplementary Fig. 15 Cytotoxicity induced by the iron chelator deferoxamine (DFO) and its viability rescue upon iron citrate supplementation, but not by sodium citrate. DFO (100 μ M) was treated to A549 cells in the absence or presence of ferric citrate (200 μ M) or sodium citrate (200 μ M) for indicated times. Cell viability is presented as % relative to the vehicle-treated cells. Data represent the mean \pm SD (n = 3). *** P < 0.001. one-way ANOVA with Dunnett's *post hoc* test.

We also agree that the expression ‘mitochondrial catastrophe’ is a bit exaggerated. Hence, we changed the title to ‘Callyspongiolide kills cells by inducing **mitochondrial dysfunction**’, and made the following changes throughout the main text as well as the supplementary information.

Main text

(page 1, title) Callyspongiolide Kills Cells by Inducing Mitochondrial Dysfunction *via* Cellular Iron Depletion

(page 2, line 26) via induction of mitochondrial dysfunction through cellular iron depletion caused by lysosomal (page 13, line 324) leading to mitochondrial dysfunction and finally cell death.

(page 15, line 368) as mitochondrial dysfunction followed by cellular iron depletion

(page 16, line 400) given that the mitochondrial dysfunction and subsequent cell death induced

(page 16, line 402) CSG kills cells by inducing mitochondrial dysfunction

(page 38, Fig. 5) leading to mitochondrial dysfunction and finally cell death.

(page 40, Fig. 6) CSG kills cells by inducing mitochondrial dysfunction via cellular iron depletion

Supplementary information

(page 1, title) Callyspongiolide Kills Cells by Inducing Mitochondrial Dysfunction *via* Cellular Iron Depletion

Reviewer #1’s comment

2. I believe it would be beneficial to show that the cells are functionally iron deficient, e.g. measure the amount of key proteins that are affected by the IRP/IRE system – transferrin receptor 1 and ferritin. I would thus suggest to perform the western blot analysis of TfR1 and FTH and show that the levels of FTH decreased while TfR1 is increased. TfR1 data on mRNA level are shown but protein is missing.

Our response

We are grateful and agree with reviewer 1’s suggestion. Western blot analysis of TfR1 and FTH1 would be helpful to strengthen that CSG induces cellular iron depletion. Indeed, FTH1 was destabilized upon CSG or DFO treatment, which was rescued by the ferric citrate supplementation, not by sodium supplementation. CSG or DFO treatment did increase TfR1 protein by 1.1 fold for 24-h treatment, less extent than its corresponding mRNA *TfR1* (by 2~3 fold, Fig. 5d). According to previous reports (*Cancer Res.* **2021**, *81*, 2289–2303.; *Oncotargets Ther.* **2019**, *12*, 4359–4377.), TfR1 expression was changed by 1.1~1.2 fold, not that significantly, and it varies among cell lines.

Main text

(page 12, line 282)

We next investigated other cellular responses to iron deficiency caused by CSG. Ferritin heavy chain (FTH1) protein and cellular heme are correlated to bioavailable cellular iron, hence, FTH1 and cellular heme level decreased by iron depletion upon CSG or DFO treatment (Supplementary Fig. 17).

(page 12, line 290)

Its corresponding protein TfR1 also showed a similar tendency to the mRNA level (Supplementary Fig. 17a) with less extent.

Supplementary information

Supplementary Fig. 17 Cellular responses to cellular iron deficiency induced by CSG treatment.

(a) Immunoblotting of FTH1, TfR1 in A549 cells upon treatment with CSG (200 nM) or DFO (100 μ M), and supplementation with or without iron citrate (200 μ M) or sodium citrate (200 μ M) for 24 h.

Reviewer #1's comment

3. Mitochondrial fragmentation is mentioned and shown on the Fig. 4a but is not quantified. I believe that using ImageJ it is possible to quantify the number of mitochondrial objects and their size and based on that the authors could obtain the quantifiable measure of mitochondrial fragmentation.

Our response

We appreciate his/her comments regarding this issue. There was a mistake in the legend of Fig. 4a–b. CSG was treated for 3 h, not 24 h, to determine TMRE staining and LysoTracker staining, hence, mitochondria were not that much fragmented shown as in Fig. 4a. To address this issue, we revised the manuscript accordingly.

Main text

(page 36, Fig. 4)

(a, b) Representative live-cell fluorescence imaging of mitochondria with active potentials using TMRE (a) and acidic lysosomes using LysoTracker (b) following treatment of A549 cells with FCCP (20 μ M), Baf (20 nM), or CSG (200 nM) for 3 h.

In fact, we already measured mitochondrial perimeter using ImageJ to quantify the mitochondrial fragmentation upon CSG treatment and the morphological restoration by iron supplementation. Please find them in Fig. 5f–g.

Rather than quantifying the images in Fig. 4a, we quantified the fragmented mitochondria in Supplementary Fig. 9b, where we first mentioned morphological change of mitochondria upon CSG treatment. Therefore, we revised Supplementary Fig. 9 and its legend.

Main text

(page 7, line 171)

The thread-like mitochondrial structures of cells were fragmented upon 12-h CSG treatment and remained so until cell death (Supplementary Fig. 9b–c).

Supplementary information

(c) Quantification of the mitochondrial perimeter in (b). Data represent the mean \pm SD (the number of quantified cells is indicated under each bar). *** $P < 0.001$, one-way ANOVA with Bonferroni's *post hoc* test.

Reviewer #1's comment

4. If the mitochondrial function is compromised and iron supplementation restores the phenotype- Is the FeS cluster and heme biosynthesis in mitochondria affected? Could authors measure heme levels by a fluorescence based method (DOI: 10.1021/ac60228a014)?

Our response

We appreciate reviewer 1's comment. Based on his/her comment, we performed the heme content assay with fluorometric method according to the reference. As shown in Supplementary Fig 17b, CSG treatment reduces heme content level, similar to DFO treatment, which was recovered by ferric citrate supplementation, but not by sodium citrate. We added the related data in the revised Supplementary Fig. 17b.

Main text

(page 11, line 272)

Ferritinophagy replenishes cellular iron and supplies it to mitochondria, where heme and iron–sulfur clusters are synthesized and utilized as cofactors for proteins in the electron-transport chain.⁴⁹

(page 12, line 283)

Ferritin heavy chain (FTH1) protein and cellular heme are correlated to bioavailable cellular iron, hence, FTH1 and cellular heme level decreased by iron depletion upon CSG or DFO treatment (Supplementary Fig. 17).

Supplementary information

Supplementary Fig. 17 Cellular responses to cellular iron deficiency induced by CSG treatment.

(b) Heme content assay with fluorometric method in A549 cells upon treatment with CSG (200 nM) or DFO (100 μ M), and supplementation with or without iron citrate (200 μ M) or sodium citrate (200 μ M) for 24 h. Relative heme content is presented as ratio to the DMSO- and DW-treated cells. Data represent the mean \pm SD (n = 3). ** $P < 0.01$, *** $P < 0.001$. one-way ANOVA with Dunnett's *post hoc* test.

(Supplementary method)

Heme content assay

Cellular heme level was measured by fluorometric method as previously published^{1,2} with minor modifications. A day after A549 cells were seeded on 6-well plate (Corning), CSG (200 nM) or DFO (100 μ M) was treated in the absence or presence of ferric citrate (200 μ M) or sodium citrate (200 μ M) for 24 h under 5% CO₂ at 37 °C. The media was aspirated, then the cells were washed with PBS. Cells were collected, then lysed in Triton X-100 lysis buffer (50 mM Tris-HCl pH 7.6, 100 mM NaCl, 1% Triton X-100, and protease-inhibitor cocktail). The soluble fraction was collected after centrifugation at 20,000 g for 20 min at 4 °C, and total protein concentration was determined using Pierce BCA protein assay kit. 1 volume of cell lysate was combined with 20 volume of supersaturated oxalic acid (about 2 M), then the samples were immediately boiled at 95 °C for 30 min. A parallel set of samples were left unheated for blank. The samples were cooled and transferred to black 96-well plate, then the fluorescence from protoporphyrin IX was measured under excitation at 400 nm and emission at 620(10) nm using Tecan Spark multimode microplate reader, then normalized to protein concentration.

Reviewer #1's comment

5. Recently, an article that described specific cell death induction by mitochondrially targeted DFO (mitoDFO) has been published (doi: 10.1158/0008-5472.CAN-20-1628). Although it induces similar output, e.g. mitochondrial dysfunction and cell death induction by iron depletion, the mechanism is apparently different, as CSG main effect seem to be the block of lysosomal acidification and also inhibition of autophagy. On the other hand the mitoDFO compound rather induces mitophagy and decreases the level of p62 and apparently lysosomal activity is retained. Could the authors comment on that and discuss it? Would they be able to assess mitophagy with the Mitophagy dye from Dojindo to define whether in CSG effect it plays some role?

Our response

We appreciate the thoughtful comment by reviewer #1. We infer that mitophagy is not involved in the cytotoxic effect upon CSG treatment. As mitophagy occurs (upon mitoDFO treatment), defected mitochondria are cleared out via autophagy process which requires acidic lysosome. The mitophagy process is featured by fragmented mitochondria, enhanced mitochondrial ROS, and decreased mitochondrial content. Upon CSG treatment, we observed fragmented mitochondria and enhanced mitochondrial ROS (Supplementary Fig. 9b, Fig. 5i), but mitochondrial contents were not changed which was confirmed by the measurement of mitochondrial DNA levels (Supplementary Fig. 9e). In addition, acidic lysosome and autophagy process are pivotal for mitophagy, but CSG impairs lysosomal acidification, which inhibits autophagy process (Fig. 4b–f). Malena *et al.* reported that EtBr-induced mitophagy showed a decreased amount of mtDNA, but co-treatment with chloroquine, an inhibitor of lysosomal acidification, blocked mitophagy process, thus mtDNA level was increased (*Autophagy*, **2016**, *12*, 2098–2112.), indicating that acidic lysosome is crucial for mitophagy process. It was reported that mitoDFO kills cells via mitophagy process mediated by mitochondrial ROS production, so mitoDFO-mediated cell death was blocked by addition of anti-oxidant NAC. However, we observed that CSG-mediated cell death was not recovered by NAC addition, which was added in the revised Supplementary Fig. 19a. Collectively, we believe that CSG kills cells independent of mitophagy, therefore we decided not to test with Mitophagy dye from Dojindo.

Main text

(page 13, line 317)

Because elevated cellular ROS can directly induce cell death as well as ROS-mediated ferroptosis,⁵⁷ we examined whether CSG cytotoxicity was associated with overloaded cellular ROS or ferroptosis. However, treatment with antioxidant *N*-acetylcysteine (NAC) or ferrostatin-1, a ferroptosis inhibitor that can prevent ferroptotic cell death via induction of RSL-3, did not rescue cells from CSG-induced death (Supplementary Fig. 19).

(page 13, line 322)

the current results indicate that CSG kills cells via depleting cellular iron induced by lysosomal dysfunction, regardless of the elevated cellular ROS or ferroptotic pathway, leading to mitochondrial dysfunction and finally cell death.

Supporting information

Supplementary Fig. 19 CSG-mediated cell death was not mediated by elevated ROS or ferroptosis.

(a) Dose-dependent cell viability of A549 cells upon treatment with CSG in the absence or presence of *N*-acetylcysteine (NAC, 1 mM) for indicated times. Cell viability is presented as % relative to the vehicle-treated cells. Data represent the mean \pm SD ($n = 3$).

Nonetheless, it is interesting that both mitoDFO and CSG similarly induce cellular iron depletion, but their detailed mechanisms underlying cell death are different. Hence, we added related content in the Discussion section as follows.

Main text

(page 15, line 369)

Maintaining iron homeostasis is crucial for cell proliferation and growth, especially, cancer cells show a high demand for iron due to their rapid proliferation.⁶⁷ Hence, targeting cellular iron homeostasis, such as the iron deprivation using CSG or DFO, could be a strategy for treating cancer, though systemic toxicity should be considered at the clinical stage. Sandoval-Acuña *et al.* introduced a mitochondria-targeting functional element to the iron chelator DFO, and named mitoDFO for selective targeting of cancer cells.⁶⁸ They showed that mitoDFO killed cells via mitophagy to clear out destabilized mitochondria upon iron depletion, and inhibited tumor growth and metastasis in a mouse model without affecting systemic iron homeostasis. Even though both mitoDFO and CSG killed cells via inducing iron depletion, our data showed that their working mechanisms are different; mitoDFO directly binds and deprives mitochondrial iron leading to mitophagy, but CSG impairs the acidification of lysosome resulting in a hampered autophagy process (Fig. 4b–f), followed by cellular iron depletion. Hence, CSG treatment did not directly induce mitophagy confirmed by unchanged mitochondrial content (Supplementary Fig. 9e), probably due to CSG-mediated lysosomal dysfunction.

Reviewer #1's comment

6. Authors provide a Seahorse analysis with OCR reduction after CSG treatment, they also show higher ROS levels after CSG, Could they assess the assembly and structure of mitochondrial respiratory supercomplexes by blue native electrophoresis (BNE)??

Our response

We do agree that the BN-PAGE experiment would be beneficial to show how CSG disrupts mitochondrial respiration. We tried to perform BN-PAGE experiments upon CSG treatment, but we were not able to obtain publishable-quality data as we are not familiar with this technique. Although we did not observe mitochondrial respiratory supercomplexes, we did obtain that some mitochondrial respiratory proteins bearing iron-sulfur cluster (NDUFS1 in Complex I, SDHB in Complex II, UQCRC1 in Complex III) were destabilized upon CSG or DFO treatment (Fig. 5e). Especially, it was reported that NDUFS1 plays a key role in the assembly of Complex I itself, and supercomplex between Complex I and Complex III (*Mol. Cell* **2019**, *74*, 452–465.e7.; *Cells* **2019**, *8*, 1149.). It was also reported that the incorporation of UQCRC1 is a penultimate step in Complex III assembly (*Mol. Cell* **2017**, *67*, 96–105.e4.; *Front. Genet.* **2015**, *6*, 134). Collectively, we indirectly inferred that these proteins (NDUFS1, SDHB, UQCRC1) were destabilized upon CSG- or DFO-mediated iron depletion, leading to the impairment of mitochondrial respiratory supercomplex and the subsequent dysfunction in cellular mitochondrial respiration. Therefore, we added the following paragraph and related references in the revised manuscript as follows.

Main text

(page 12, line 299)

Especially, NDUFS1 and UQCRC1 have a key role in the assembly of mitochondrial respiratory complex,^{52–55} hence, such destabilization of the proteins upon CSG or DFO treatment could lead to the impaired mitochondrial respiration.

52. Elkholi, R. *et al.* MDM2 integrates cellular respiration and apoptotic signaling through NDUFS1 and the mitochondrial network. *Mol. Cell* **74**, 452–465.e7 (2019).

53. Ni *et al.* Mutations in NDUFS1 cause metabolic reprogramming and disruption of the electron transfer. *Cells* **8**, 1149 (2019).

54. Bottani, E. *et al.* TTC19 plays a husbandry role on UQCRC1 turnover in the biogenesis of mitochondrial respiratory complex III. *Mol. Cell* **67**, 96–105.e4 (2017).

55. Fernández-Vizarra, E. & Zeviani, M. Nuclear gene mutations as the cause of mitochondrial complex III deficiency. *Front. Genet.* **6**, 1–11 (2015).

Reviewer #1's comment

7. In general, the paper is very interesting but the question is whether there are data that would suggest that the observed effect is specific to cancer cells and does not affect for example non-malignant normal human cells such as fibroblasts? Are there any data on its in vivo effect and toxicity? I have not found such studies but the authors could correct me and should discuss this as well.

Our response

We appreciate the thoughtful comment by reviewer #1. In our study, we clearly showed that CSG inhibits autophagic flux and disrupts cellular iron homeostasis. Our data showed that CSG inhibits cancer cell proliferation, but CSG can also affect the viability of normal cells on the basis of its mechanism of action. Autophagy inhibition and impaired iron homeostasis could lead to systemic toxicity at the organism level. Indeed, we performed a cell viability assay of CSG in MRC-5 human normal lung fibroblast cells and confirmed that CSG induces cytotoxicity even at 24-h treatment (**Review Figure 1**). IC_{50} at 72-h treatment in MRC-5 cells was 88.7 ± 3.8 nM, which was slightly higher than that of A549 cells (64.8 ± 1.1 nM). Hence, *in vivo* study with CSG might be difficult to apply. CSG is not a cancer-specific cytotoxic natural product, therefore specific delivery of CSG to cancer cells might be a potential strategy for the development of anti-cancer agents. In fact, CSG has been applied as a warhead of antibody-drug conjugate for selective delivery.

Reviewer Figure 1. Dose-dependent cell viability of MRC-5 normal lung fibroblast cells following treatment with CSG for indicated times. Cell viability is presented as % relative to the vehicle-treated cells. Data represent the mean \pm SD (n = 4).

Reviewer #2's comment

This manuscript reveals that the cytotoxicity of callyspongiolide (CSG) on cancer cells is dependent on the induction of mitochondrial catastrophe through cellular iron depletion caused by lysosomal deacidification. Of note, TS-FITGE, a label-free target identification method, was employed in discovering ten possible direct protein targets of CSG. The manuscript is clearly written, but some revisions are required to improve the overall quality:

We appreciate the reviewer's strong support on our study along with his/her thoughtful suggestions and comments.

Reviewer #2's comment

1) Previous study reports that *epi*-CSG is more potent than the natural form of CSG in killing cancer cells. The authors may explain why CSG is more potent in this study.

Our response

Zhou *et al.* reported that *epi*-CSG was more potent than natural form of CSG (CSG) in MCF7, SH-SY5Y, HeLa, HT-29, RKO, PC-3 cells, and *vice versa* in H1299, Jurkat cells (*J. Am. Chem. Soc.* **2016**, *138*, 6948–6951). However, we herein reported that CSG is more potent than *epi*-CSG in all cell lines we tested (A549, HCT116, HEK293T, HeLa, HepG2, Jurkat, MCF7, PC-3, SH-SY5Y), which is consistent with very recent paper reported by Lee *et al.* showing that CSG was more potent than *epi*-CSG in SKOV3, N87, HT29, HPAFII, SW480, MCF7, HCT116, A549, HepG2, 786-O (*BMB Rep.* **2021**, *54*, 227–232). We also confirmed the optical rotation of CSG and *epi*-CSG for their authenticity. We obtained $[\alpha]_D^{20}$ (*c* 0.1, MeOH) = -12.17 for CSG and -52.85 for *epi*-CSG, that are similar to the reported values, -12.5, -13.0 for CSG and -62.86 for *epi*-CSG in several reports (*Org. Lett.* **2014**, *16*, 266–269.; *J. Am. Chem. Sci.* **2016**, *138*, 6948–6951). Considering these observations, we infer that there were some mistakes in the previous study.

Reviewer #2's comment

2) “CSG-mediated cytotoxicity was unaffected by PP2AC depletion” is not accurate, because observable expression of PP2AC still appeared after siRNA treatment as the western blots show. The authors may knockdown other 9 proteins identified by TS-FITGE to see whether it affects CSG activity.

Our response

We appreciate the thoughtful comment by reviewer #2. To address this issue, we purchased several siRNAs for PP2AC from several vendors, and optimized the lipofectamine-based transfection condition without cytotoxicity. However, the complete depletion of PP2AC was not obtained. Therefore, we changed the expression ‘PP2AC depletion’ to ‘PP2AC knockdown’ in the revised manuscript as well as the supplementary information.

Main text

(page 7, line 150) but CSG-mediated cytotoxicity was unaffected by PP2AC knockdown in the cells (Supplementary Fig. 7).

Supplementary information

(page 9, Supplementary Fig. 7)

CSG-mediated cell-death potency was not altered upon PP2AC knockdown in the cells.

(b) Dose-dependent cell viability upon treatment of CSG for indicated times in A549 cells upon PP2AC knockdown.

The reason why we selected PP2AC for siRNA treatment study is that only PP2AC showed the thermal stability shift upon CSG treatment, which means that the physical engagement between PP2AC and CSG was confirmed by CETSA experiments (Supplementary Fig. 6). Hence, we did not proceed to siRNA treatment studies with other proteins because their associations with CSG were not confirmed in CETSA experiment.

Reviewer #2's comment

3) "... protein spot separation was not relevant to the function of the electron-transport chain". The conclusion is contradictory to the fact that 10 protein spots were identified by TS-FITGE, belonging to the mitochondrial electron-transport system.

Our response

We do agree that the phrase is ambiguous. Although most of proteins obtained from TS-FITGE experiment belong to mitochondrial respiratory complex, the protein spot separation on 2-D gels at high temperatures still remained in the presence of respiratory complex functional inhibitor including rotenone, antimycin A, and oligomycin A (Supplementary Fig. 10). Therefore, we revised the manuscript as follows.

Main text

(page 8, line 183)

TS-FITGE performed with CSG experiments following pretreatment with specific complex I, III, and V inhibitors [rotenone (Rot), antimycin A (AA), and oligomycin A (Oligo), respectively] revealed the continued existence of green-red spot pairs at high temperature (Supplementary Fig. 10a–d), indicating that protein spot separation did not occur through perturbation of the mitochondrial respiratory function even though most of the proteins obtained from TS-FITGE experiments belong to the electron-transport chain complex.

Reviewer #2's comment

4) "This observation implied that FCCP treatment phenocopied the TS-FITGE outcomes for CSG". Are these proteins the same as that identified from CSG treated cells? The authors may employ tandem mass spectroscopy to characterize them.

Our response

We empirically knew that proteins in certain 2-D gel spots are identical if their positions on 2-D gels are superimposable. Although the vertical position of Region 1–4 on 2-D gels could be different among each experiment due to acrylamide percentage in each gel, we can confirm that the Region 1–4 were all identical in all TS-FITGE experiments (TS-FITGE of CSG alone or CSG pretreated with rotenone, antimycin A, oligomycin A, or FCCP, TS-FITGE of FCCP, TS-FITGE of bafilomycin A1, and TS-FITGE of chloroquine) by comparing the whole spot patterns on 2-D gels. Therefore, we can track changes in each spot of 2-D gels by directly comparing those spots using Melanie software. To address the reviewer's comment, we revised "Region 1 in Figure 3d" to show that this region is identical to other "Region 1" images.

Reviewer #2's comment

6) "CSG was slightly less toxic under ATG5 depletion". Western blots result indicates that there is still some expression of ATG5 after siRNA treatment.

Our response

We appreciate the thoughtful comment by reviewer #2. To address this issue, we purchased several siRNAs for ATG5 from several vendors, and optimized the lipofectamine-based transfection condition without cytotoxicity. However, the complete depletion of ATG5 was not obtained. Therefore, we changed the expression 'ATG5 depletion' to 'ATG5 knockdown' in the revised manuscript as well as the supplementary information.

Main text

(page 10, line 245) Following the knockdown of autophagy-related 5 (ATG5)

(page 10, line 249) CSG was slightly less toxic under ATG5 knockdown

Supplementary information

(page 17, Supplementary Fig. 14)

CSG-mediated cell death was not significantly affected upon ATG5 knockdown.

(b) Dose-dependent cell viability following treatment with CSG for indicated times in A549 cells upon ATG5 knockdown.

Reviewer #2's comment

7) In Fig. 5 b-c, the decreasing immunofluorescence maybe caused by the cytotoxicity of staining. The authors should use better controls.

Our response

FerroOrange is an iron-sensitive fluorescent dye. During 24-h treatment of either DFO or CSG, we observed no cytotoxic effect in A549 cells. Please find that cell viability of A549 cells was not affected by 24 h-treatment of CSG, shown in Supplementary Figure 1. We also added cell viability data upon DFO treatment in the revised Supplementary Fig. 15, upon suggestion by Reviewer 1's Comment #1, and brightfield images of FerroOrange-staining experiments in the revised Supplementary Figure 16 to show the overall cell morphology. In addition, we revised FerroOrange images in Figure 5b and Supplementary Figure 16 to improve the clarity of images.

Main text

(page 12, line 279)

FerroOrange fluorescence was enhanced as cellular iron was replenished via ferric citrate supplementation, but diminished as a result of cellular iron depletion using DFO without cytotoxicity for 24-h treatment (Supplementary Fig. 15).

Supplementary information

Supplementary Fig. 16 Representative A549 live-cell fluorescent images with FerroOrange staining and brightfield images upon treatment of ferric citrate (Fe, 200 μ M), deferoxamine (DFO, 100 μ M) for 24 h, or CSG (200 nM) for indicated times. Scale bar, 10 μ m.

Reviewer #2's comment

8) It would be helpful if the authors predict possible target(s) of CSG in killing cancer cells.

Our response

We pursued TS-FITGE technique to identify direct binding partner(s) of CSG, but we failed to find the proteins whose thermal stability shifts were altered upon CSG treatment except PP2AC. In fact, PP2AC was functionally irrelevant to cell death caused by CSG (Supplementary Fig. 7). Instead, we observed a unique phenomenon of protein spot separation on 2-D gels at increasing temperatures, which revealed the association of CSG to dysregulated lysosomal acidity. Although we were not able to fulfill the original goal for identifying functional target protein(s) of CSG, we cautiously speculated possible target proteins of CSG as certain subunits of vacuolar-type ATPases including ATP6V0C on the basis of our own data and a previous report by Forgarty *et al.* Therefore, we added the following paragraph in the Discussion section of the revised manuscript as follows.

Main text

(page 14, line 335)

Though TS-FITGE did not give information about the direct targets of CSG, we cautiously speculate that CSG would bind to and inhibit certain subunits of vacuolar-type ATPases including ATP6V0C that can lead to the inhibition of lysosomal acidification, supported by the inhibition of vacuolar-type ATPase activity upon CSG treatment, similar to bafilomycin A1 treatment (Fig. 4c), and a previous study reporting vacuolar-type ATPase as a potent target of CSG in yeast.¹⁷

Reviewer #2's comment

9) In the discussion part: "The results confirmed the applicability of TS-FITGE for the ID of direct target proteins" is not accurate, as the authors did not find direct targets of CSG using TS-FITGE in this manuscript.

Our response

We do agree with the reviewer's comment. The former sentence can mislead the audience. Therefore, we revised the manuscript accordingly.

Main text

(page 16, line 391)

Our results confirmed the applicability of TS-FITGE for monitoring the alteration of cellular environments upon compound treatment on the basis of pattern changes in thermal stability of protein spots, along with its original goal in target identification.

Reviewer #3's comment

1. The current study offered new insights for the mode of action of the natural product callyspongiolide (CSG). The authors confirmed the antitumor activities of CSG and tried to identify the target protein. Moreover, they found that CSG is a potent inhibitor of the Vacuolar ATPase in mammalian cells, which is consistent with findings in yeast (ref. 17). In addition, they found iron depletion by CSG leads to cell death. These important novel findings are surely interesting to readers of this journal and draw more discussions in the field.

Our response

We deeply appreciate the reviewer for this encouraging comment to our works.

Reviewer #3's comment

2. This reviewer is not familiar with methods for label-free target identification, especially TS-FITGE (thermal stability shift-based fluorescence difference in two-dimensional (2-D) gel electrophoresis). No further comments on those experiments involving TS-FITGE should be given. But it's clear that this method has been mentioned quite often in the manuscript and used initially to look for protein target of CSG and eventually led the authors "questioned whether CSG affects mitochondrial morphology and respiratory function". Maybe it's appropriate they add TS-FITGE to the title.

Our response

We appreciate reviewer 3's suggestion and strong support on our study, but we don't think that using TS-FITGE in the title is appropriate. The TS-FITGE method was mentioned several times in the main text, but the core finding was that CSG kills cells by inducing lysosomal dysfunction that leads to mitochondrial dysfunction via iron depletion. In this study, we used TS-FITGE as a chemical biology research tool. Furthermore, the name TS-FITGE may not be familiar to general readers, therefore we decided not to include TS-FITGE in the title of the revised manuscript.

Reviewer #3's comment

3. The current manuscript described experiments supporting CSG as an inhibitor of Vacuolar ATPase in HEK293T cells. This confirms that CSG is a potent inhibitor of the Vacuolar ATPase (in yeast, ref. 17). Then, abnormal lysosome acidity seems to be the natural result of inhibiting Vacuolar ATPase. But the authors went to TS-FITGE data for clues. That looks like an unnecessary detour.

Our response

We appreciate the thoughtful comment by reviewer 3. While we pursued the target identification of CSG using TS-FITGE to reveal its mechanism of action, the paper was published revealing CSG as a potent inhibitor of the vacuolar ATPase (*J. Nat. Prod.* **2020**, *83*, 3381–3386). However, in our study, we observed the unique event using TS-FITGE, the protein spot separation on the 2-D gels at increasing temperatures as the lysosomal acidity is impaired. This observation using TS-FITGE provided us a clue to dissect the detailed mechanism of action, which might be useful for MOA study of other bioactive small molecules.

Reviewer #3's comment

4. The authors concluded that CSG kills cells by inducing mitochondrial catastrophe via cellular iron depletion, based on FerroOrange imaging and ferric citrate supplementation experiments. As compared to the view of energy deprivation induced by mitochondrial dysfunction (ref. 62), the current study offered a different mechanism with new evidences, although no further discussions about alternatives were found in the manuscript. I am thinking of the report by Hughes et al. (*Cell*, 2020, p296-310)-they pointed out "cysteine depletion or iron supplementation restores mitochondrial health in vacuole-impaired cells and prevents mitochondrial decline during aging".

Our response

We are grateful for reviewer 3's suggestion. Obviously, reviewer 3 is an expert and well-versed in the field. We found that Hughes *et al.* revealed that dysfunctional acidic vacuoles deprive cellular iron through ROS-dependent mechanism in yeast, which was different from what we observed in mammalian cells. Therefore, we added the following paragraph in the Discussion section of revised manuscript as follows.

Main text

(page 16, line 382)

As noted, the acidic endosome and lysosome have pivotal roles in supplying bioavailable iron through endocytosis of ferritins and ferritinophagy. Hughes *et al.* observed that dysfunctional acidic vacuoles in yeast can be a major cause of age-related mitochondrial deterioration through ROS-dependent iron depletion.⁶⁹ However, we observed that CSG-mediated cell death was not rescued by the addition of NAC (Supplementary Fig. 19), and the iron supplementation alleviated CSG-induced ROS generation (Fig. 5i). These results indicated that CSG treatment in mammalian cells resulted in cellular iron deprivation, leading to the impairment of mitochondrial respiration and subsequent ROS generation, rather than ROS generation followed by cellular iron depletion reported in yeast.

REVIEWERS' COMMENTS:

Reviewer #1 (Remarks to the Author):

The authors have addressed my comments and concerns adequately, I do not have further comments.

Reviewer #2 (Remarks to the Author):

I thank the authors for addressing my comments.